# Sporulation Activated via $\sigma^W$ Protects *Bacillus* from a Tse1 Peptidoglycan Hydrolase Type VI Secretion System Effector

Alicia I. Pérez-Lorente,[a] Carlos Molina-Santiago,[a] Antonio de Vicente,[a] Diego Romero[a]

[a]Instituto de Hortofruticultura Subtropical y Mediterránea La Mayora, Universidad de Málaga-Consejo Superior de Investigaciones Científicas, Departamento de Microbiología, Universidad de Málaga, Málaga, Spain

**ABSTRACT** Within bacterial communities, community members engage in interactions employing diverse offensive and defensive tools to reach coexistence. Extracellular-matrix production and sporulation are defensive mechanisms used by *Bacillus subtilis* cells when they interact with *Pseudomonas chlororaphis* strains expressing a type VI secretion system (T6SS). Here, we define Tse1 as the main toxin mobilized by the *Pseudomonas chlororaphis* T6SS that triggers sporulation in *Bacillus subtilis*. We characterize Tse1 as a peptidoglycan hydrolase that indirectly alters the dynamics and functionality of the *Bacillus* cell membrane. We also delineate the response of *Bacillus* cells to Tse1, which through the coordinated actions of the extracellular sigma factor $\sigma^W$ and the cytoplasmic histidine kinases KinA and KinB, culminates in activation of the sporulation cascade. We propose that this cellular developmental response permits bacilli to defend against the toxicity of T6SS-mobilized Tse1 effector.

**IMPORTANCE** The study of bacterial interactions is helping to define species-specific strategies used to modulate the competition dynamics underlying the development of community compositions. In this study, we deciphered the role of *Pseudomonas* T6SS when competing with *Bacillus* and the mechanism by which a T6SS-toxin modifies *Bacillus* physiology. We found that *Pseudomonas* triggers *Bacillus* sporulation by injecting through T6SS a toxin that we called Tse1. We found that Tse1 is a hydrolase that degrades *Bacillus* peptidoglycan and indirectly damages *Bacillus* membrane functionality. In addition, we demonstrated the mechanism by which *Bacillus* cells increase the sporulation rate upon recognition of the presence of Tse1. Interestingly, asporogenic *Bacillus* cells are more sensitive to T6SS activity, which led us to propose sporulation as a last resort of bacilli to overcome this family of toxins.

**KEYWORDS** sigma factors, T6SS, *Bacillus subtilis*, *Pseudomonas fluorescens*, interspecies interaction, sporulation

**M**ultispecies microbial communities are constantly in competition for scarce resources, such as space and nutrients (1). A diverse battery of offensive and defensive tools, including secretion of compounds (e.g., siderophores or antibiotics) and extracellular-matrix production, are deployed to modulate inter- and intraspecies coexistence (2–5). An outstanding example of cellular machinery implicated in shaping microbial community structure is the type VI secretion system (T6SS), a molecular nanomachine designed to inject effectors into competitor cells (6). Initially described in *Vibrio cholerae* (7) and *Pseudomonas aeruginosa* (8), the T6SS is now believed to be present in 25% of Gram-negative bacteria. T6SSs are composed of 13 core components that form a versatile bacteriophage-like structure. The integral membrane complex is formed by TssJ, TssM, and TssL, while the baseplate is formed by TssE, TssF, TssG, and TssK. The TssB and TssC subunits are responsible for assembly of the contractile sheath, which is formed by hexamers of Hcp (9–11). Another core component of T6SS is TssA, an essential cap protein involved in coordinating tail polymerization (12). The tip of

Address correspondence to Diego Romero, diego_romero@uma.es.

The authors declare no conflict of interest.

the spike is formed by a trimer of VgrG, a valine-glycine repeat protein, and PAAR, a proline-alanine-alanine-arginine repeat protein that facilitates delivery of specific effectors (13–15). Finally, the ClpV ATPase is involved in the disassembly and recycling of T6SS components (16–18) (see Fig. S1A and B in the supplemental material).

T6SSs of various microbes have been reported to mobilize effectors against other Gram-negative bacteria (19) and/or eukaryotic cells during bacterial infection (6, 20, 21). Recently, it has been demonstrated that *Serratia marcescens* expresses a T6SS that secretes two effectors capable of acting against fungal cells (22). Two additional studies have reported effects of T6SSs against Gram-positive bacteria. Among pseudomonads, *Pseudomonas chlororaphis* PCL1606 (referred to here as Pchl), a soil-dwelling bacterium known for its biocontrol activity against avocado pathogens, was recently reported to encode a T6SS that triggers sporulation in the Gram-positive bacteria *Bacillus subtilis* upon close cell-to cell contact (4, 23). Furthermore, *Acinetobacter baumannii* has been shown to mobilize a bifunctional effector that seems to kill *B. subtilis* and other Gram-positive bacteria (24).

*Bacillus subtilis* NCIB3610 (referred to here as Bsub) is a Gram-positive soil dwelling bacterium frequently used as a model in studies of bacterial cell differentiation and gene expression regulation (25–27). In ecologically relevant environments, Bsub self-defense relies on at least two complementary mechanisms: (i) the production of an extracellular matrix, which protects bacterial colonies from toxic compounds produced by competitors, and (ii) the formation of spores in response to external cues that threaten bacterial cell viability (4). Initiation of the sporulation pathway is orchestrated by the coordinated action of kinases that ultimately increase the intracellular level of the phosphorylated form of the master regulator Spo0A, which then activates a well-understood genetic cascade (28–30). Briefly, the histidine kinases KinA and KinB are the first ones responsible for initiation of the sporulation cascade upon phosphorylation of Spo0F. After that, Spo0F phosphorylates Spo0B, which passes the phosphate on to Spo0A. Kinases KinC and KinD also control initiation of sporulation, being able to act directly on Spo0A (30, 31). *Pseudomonas* spp. and *Bacillus* spp. are among the most studied beneficial soil microbes found in rhizosphere communities (32, 33). These species can establish synergistic and antagonistic relationships; however, these relationships have scarcely been analyzed, and it is still necessary to study the implications of the coexistence between these two genera (5, 34–36).

In a previous study, we observed that Bsub cells sporulated in the presence of Pchl strain expressing a T6SS$^{Pchl}$ (4). In this work, we identified and elucidated the mechanism by which a T6SS$^{Pchl}$ effector, Tse1, modifies Bsub bacterial anatomy and physiology to trigger sporulation. We additionally show that Bsub responds to the Tse1 offensive via the coordinated activities of the extracytoplasmic sigma factor $\sigma^W$ and the cytoplasmic histidine kinases KinA and KinB, which ultimately result in sporulation-triggering phosphorylation.

## RESULTS

**The T6SS cluster possesses several putative toxins.** T6SSs have been established as an effective bacterial appendage that modulates inter- and intraspecies bacterial interactions in a spatially dependent manner. We were interested in deciphering the exact mechanism underlying the interaction between Bsub and Pchl with an active T6SS. An *in silico* analysis of the Pchl genome (37) revealed a complete T6SS gene cluster of 27.9 kb (T6SS$^{Pchl}$) that is similar in structure and composition to the T6SSs of other soil-dwelling bacteria, including *Pseudomonas syringae* pv. tomato DC3000, *Pseudomonas aeruginosa* PAO1, and *Pseudomonas putida* KT2440 (Fig. 1A). A phylogenetic analysis based on the *tssB* locus positioned T6SS$^{Pchl}$ in group 1.1, placing it very close to the HIS-I T6SS of *P. syringae* pv. tomato DC3000, H2 T6SS of *P. aeruginosa* PAO1, and K2 T6SS of *P. putida* KT2440 (Fig. 1B). The T6SS$^{Pchl}$ main gene cluster comprises (i) 12 structural protein components of the T6SS syringe; (ii) two putative T6SS effectors based on BLASTP (38) and the SecReT6 database (39), one linked to a *paar*

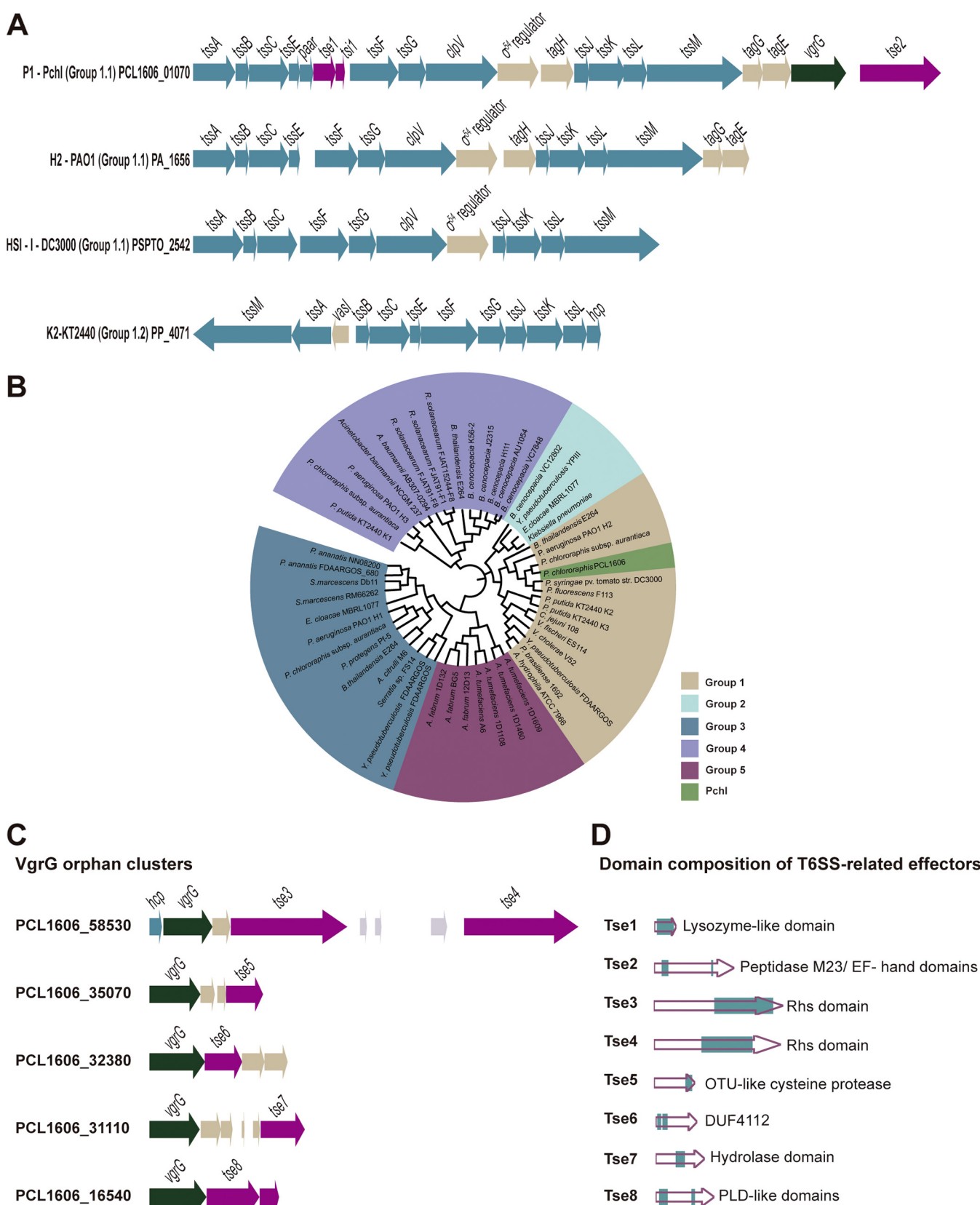

**FIG 1** Characterization of the *P. chlororaphis* T6SS. (A) Genetic architecture of various *Pseudomonas* group 1 T6SS clusters. Schematic representations of the main Pchl T6SS cluster, the H2 T6SS cluster of *P. aeruginosa* PAO1, the HS1-1 T6SS of *P. syringae* pv. tomato DC3000, and the K2 T6SS of *P. putida* KT2440. (B) Phylogenetic distribution of T6SS clusters in five main groups based on the *tssB* locus. Phylogenetic analysis was done using previous nomenclature (107, 108) based on the core protein TssB, widely used as structural component for T6SS phylogenetic analysis (33, 109). The maximum-likelihood method

gene and the other to a *vgrG* gene; and (iii) regulatory sequence for the sigma factor $\sigma^{54}$, which is involved in differential regulation of T6SSs in *Pseudomonas aeruginosa*, *Salmonella enterica*, *Vibrio cholerae*, and *Aeromonas hydrophila* (40, 41). We additionally identified six other putative T6SS effectors encoded in five orphan *vgrG* islands (Fig. 1C).

We further analyzed the domain architecture of the eight putative effectors using Pfam and Phyre2 (Fig. 1D). Tse1 (type VI secretion system effector) is encoded in the main gene cluster and is linked to a *paar* gene and followed by its putative cognate immunity protein Tsi1 (type VI secretion system immunity protein) (toxin-immunity pair). Immunity proteins are encoded downstream of the effectors, and they protect cells both from intoxication by adjacent sibling cells and from self-intoxication (42, 43). Tse1 is a hypothetical protein with predicted lysozyme-like activity based on amino acid sequence homology. Tse2, also encoded by a gene in the main gene cluster, is a putative peptidase of the M23 family, with a C-terminal EF hand domain. Tse3 and Tse4 genes are located in the same *vgrG* orphan cluster, and both contain a C-terminal Rhs (rearrangement hot spot) domain. Effectors containing these domains usually target nucleic acids in the target cells (44, 45). Upstream of *vgrG* is *hcp*, which encodes a structural component essential for T6SS function. Tse5 is a putative cysteine protease, likely having a function similar to that of an effector recently described in *Edwardsiella piscicida* (46). Tse6 contains two uncharacterized DUF4112 domains in its N terminus, similar to those found primarily in *Acinetobacter* strains. Tse7 contains a conserved hydrolase domain. Tse8 is a putative phospholipase with two PLD (phospholipase D)-like domains and is associated with the Tsi8 immunity protein. We did not identify genes encoding immunity proteins for the Tse3, Tse4, Tse5, Tse6, and Tse7 putative effectors.

**Tse1 is a T6SS effector that triggers *B. subtilis* sporulation.** To validate our *in silico* analysis by verifying the functionality of T6SS$^{Pchl}$, we studied the Bsub sporulation percentage in competitive experiments with Pchl strains carrying mutations in different T6SS$^{Pchl}$ components. Bsub sporulation percentage increased up to 80% in the presence of the WT Pchl strain, while it decreased by 40% upon interaction with a *tssA* mutant compared to sporulation with the wild-type (WT) Pchl strain (Fig. S2A). No significant difference was observed in total CFU during coculture of strains (Data Set S8). TssA was recently shown to be essential for the assembly and polymerization of the T6SS sheath in *P. aeruginosa* and *Escherichia coli*; thus, *tssA* mutants lack active T6SSs (47–49). Accordingly, the WT Pchl strain but not the *tssA* mutant strain killed an *E. coli* biosensor strain constitutively expressing green fluorescent protein (GFP) (50). This observation supports the existence of a functional T6SS in Pchl that attacks Bsub. *P. putida* KT2440 (51), which possesses three T6SSs, two of them (K2 T6SS and K3 T6SS) phylogenetically close to Pchl, failed to induce Bsub sporulation (Fig. S2A), although it retained toxicity against *E. coli* (Fig. S2B). As expected, the triple mutant, KT2440ΔT6SS, failed to kill *E. coli*. These findings support the existence of a functional T6SS in Pchl that attacks Bsub.

The failure of T6SS$^{KT2440}$ to induce Bsub sporulation or death suggested that Bsub sporulation might be triggered by a specific T6SS effector encoded in the Pchl genome rather than by physical damage inflicted by the penetration of the syringe. Additional evidence also supported this hypothesis: (i) a strain mutant for *hcp*, which encodes an essential structural component for the assembly and contraction of the tube, failed to trigger sporulation or to kill *E. coli*, and (ii) a strain mutant for *paar*, which encodes a structural component of the spike of certain strains but is not essential for tube

**FIG 1** Legend (Continued)

with 1,000 bootstrap replicates was built with Mega X (110) using Clustal Omega (97). The final representation was built using Intel (98). Group 1 is represented in brown, group 2 in light blue, group 3 in dark blue, group 4 in purple, and group 5 in pink. Pchl (green) belongs to group 1. (C) Schematic representation of the VgrG orphan clusters of Pchl. (D) Schematic representation of the predicted domains of eight putative T6SS effectors. Genes encoding structural T6SS proteins are in blue, transcriptional genes in brown, adaptor genes in light brown, VgrG genes in green, and effectors/immunity genes in pink. All arrows representing T6SS genes and the domain sizes of putative effectors are to scale. Full names and accession numbers of strains used to generate T6SS phylogenetic tree in panel B are available in Data Set S1. In addition, Data Set S2 shows the output of a BLASTP analysis comparing the Tse1 amino acid sequence against the NCBI database.

assembly, failed to trigger Bsub sporulation but still killed *E. coli* (Fig. 2A and B). PAAR has been reported to facilitate the mobilization of specific effectors normally encoded in proximal downstream loci (14). Our *in silico* analysis revealed that *tse1*, which encodes a hypothetical protein with a putative lysozyme domain, is located immediately downstream of *paar* (Fig. 1A). A *tse1* mutant failed to induce Bsub sporulation but killed

*E. coli*. Based on these findings, we concluded that Tse1 is a T6SS$^{Pchl}$-mobilized effector that is involved in triggering Bsub sporulation.

Inspired by this finding, we then purified Tse1 to homogeneity and evaluated its activity against Bsub (Fig. S2C and D). A 7 $\mu$M solution of purified Tse1 significantly increased the Bsub sporulation percentage (43%) compared with that of untreated cells (Fig. 2C). Tse1 added directly to Bsub cells retained activity; however, this was lower (43% sporulation of Bsub) than that of Tse1 injected by T6SS of the WT strain (80% sporulation of Bsub). This finding confirmed that effectiveness of Tse1 is increased upon injection by T6SS. T6SS effectors are usually encoded near their cognate immunity proteins, and both are concurrently produced to avoid self-intoxication (52). *tsi1* is located immediately downstream of *tse1* (Fig. 1A); thus, we speculated that these proteins might form an effector-immunity (EI) pair. No significant differences in the sporulation percentage of Bsub cells overexpressing the EI protein Tsi1 (Bsub+pDR111-*tsi*) were observed in competition assays with WT Pchl (functional T6SS) or a $\Delta hcp$ (nonfunctional T6SS) strain, indicating that Tsi1 produced in Bsub cells can neutralize the toxic effect of Pchl-secreted Tse1 (Fig. 2D).

**Tse1 is a hydrolase that degrades *B. subtilis* peptidoglycan.** We speculated that the cell wall is the main target of Tse1 for two reasons: (i) the predicted lysozyme domain in Tse1 and (ii) the extracellular toxicity of Tse1 to Bsub cells. Transmission electron microscopy of thin sections of Bsub cells after incubation with Tse1 for 3 h revealed abundant ghost cells, residual bacterial cells in which leakage of intracellular contents had occurred (Fig. 3A). Complementary transmission electron microscopy confirmed the cell wall defects in Bsub cells (Fig. S3). Immunofluorescence assays showed that His-tagged Tse1 colocalized with Alexa Fluor 647-conjugated wheat germ agglutinin (WGA), a fiducial marker of local cell wall growth that binds *N*-acetylglucosamine residues at the septa and poles of Bsub cells (Fig. 3B; Fig. S4A and B) (53–55).

To further validate the hypothetical enzymatic activity of Tse1, purified Bsub sacculi were treated with Tse1 and the products were analyzed via liquid chromatography-mass spectrometry (LC-MS). Consistent with the suspected putative enzymatic activity, the elution profile of muropeptides was similar to that obtained when lysozyme was used as a control (Fig. 3C). In addition, two specific peaks were more abundant: 371.23 Da, corresponding to the disaccharide tetrapeptide NAG (*N*-acetylglucosamine)-NAM (*N*-acetylmuramic acid)-Ala-Glu-mDAP (*meso*-diaminopimelic acid)-Ala, and 415.25 Da, corresponding to the disaccharide tripeptide NAG-NAM-Ala-Glu-mDAP (Fig. 3D and Fig. S5) (56–58). Altogether, these findings confirmed the peptidoglycan hydrolase activity of Tse1, which might be a member of the muramidase enzyme family.

The damage inflicted on the peptidoglycan led us to analyze the integrity and functionality of the cell membrane. First, staining with propidium iodide (PI), a DNA intercalator used to evaluate cell membrane integrity (59), revealed a significant increase in the number of permeable Bsub cells after treatment with purified Tse1 (Fig. 4A and Fig. S6A). Second, staining with TMRM (tetramethylrhodamine, methyl ester), a widely used dye to evaluate membrane potential in bacteria (60, 61), showed a reduction of the mean fluorescence intensity, indicating a loss of membrane potential (Fig. 4B and Fig. S6B). Third, staining with DilC12 (tetramethylindocarbocyanine perchlorate), used to study membrane fluidity in bacteria (62, 63), revealed the presence of highly fluorescent foci (increased fluidity), indicating that Tse1 increased the fluidity of the cell membrane (Fig. 4C).

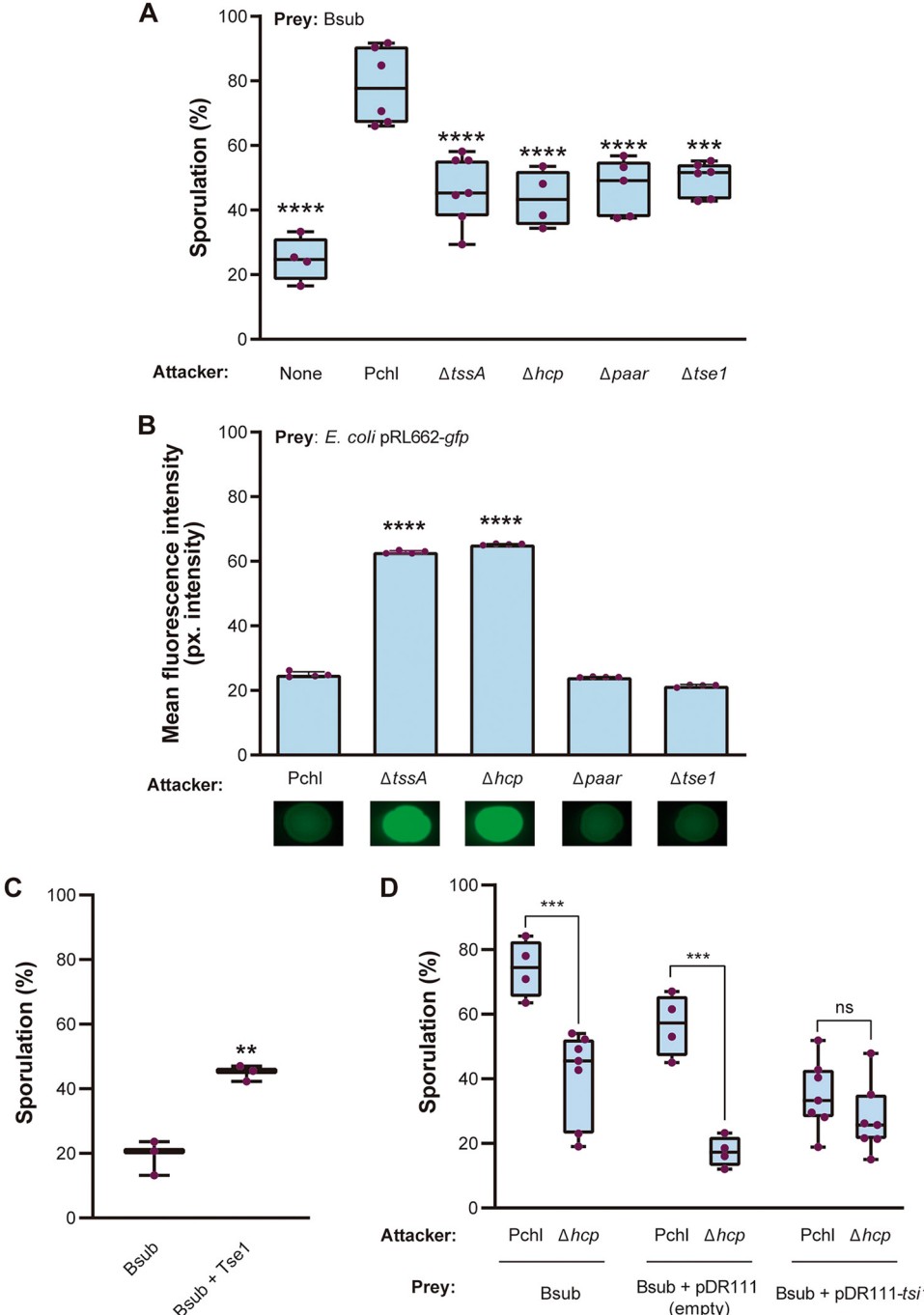

**FIG 2** Tse1 is a toxic effector capable of triggering *Bacillus* sporulation. (A) Competition assays. Quantification of the percentage of sporulated Bsub cells during competition with different Pchl strains as attackers (Pchl and the Δ*tssA*, Δ*hcp*, Δ*paar*, and Δ*tse1* mutants). (B) T6SS killing assays with *E. coli* as an indicator. Fluorescence signal measurements from *E. coli* pRL662-gfp after incubation with different Pchl strains (Pchl and the Δ*tssA*, Δ*hcp*, Δ*paar*, and Δ*tse1* mutants). Images show a representative spot of each assay after 24 h of incubation. (C) Quantification of the percentage of sporulated Bsub cells after treatment with purified Tse1 for 3 h. (D) Competition assays of Pchl (functional T6SS) or the Δ*hcp* mutant (nonfunctional T6SS) against the preys *Bacillus* (WT) and Bsub pDR111-tsi1 expressing the putative cognate immunity protein Tsi1 under the control of a constitutive promoter. In all represented competition assays (A, B, and D), the attacker and prey strains were cultured at a ratio of 1:1. After 24 h, Bsub cells were serially diluted and placed in LB for estimation of cell density (CFU) sporulation percentages. For all experiments, at least three biological replicates are shown, and the error bars represent SD. Statistical significance was assessed via *t* tests and one-way ANOVA followed by Dunnett tests using sporulation of Bsub when competing with Pchl as the control group. **, $P < 0.01$; ***, $P < 0.001$; ****, $P < 0.0001$; ns, not significant. Uncropped images of gel electrophoresis and Western blots of Tse1-tagged protein are shown in Data Set S9.

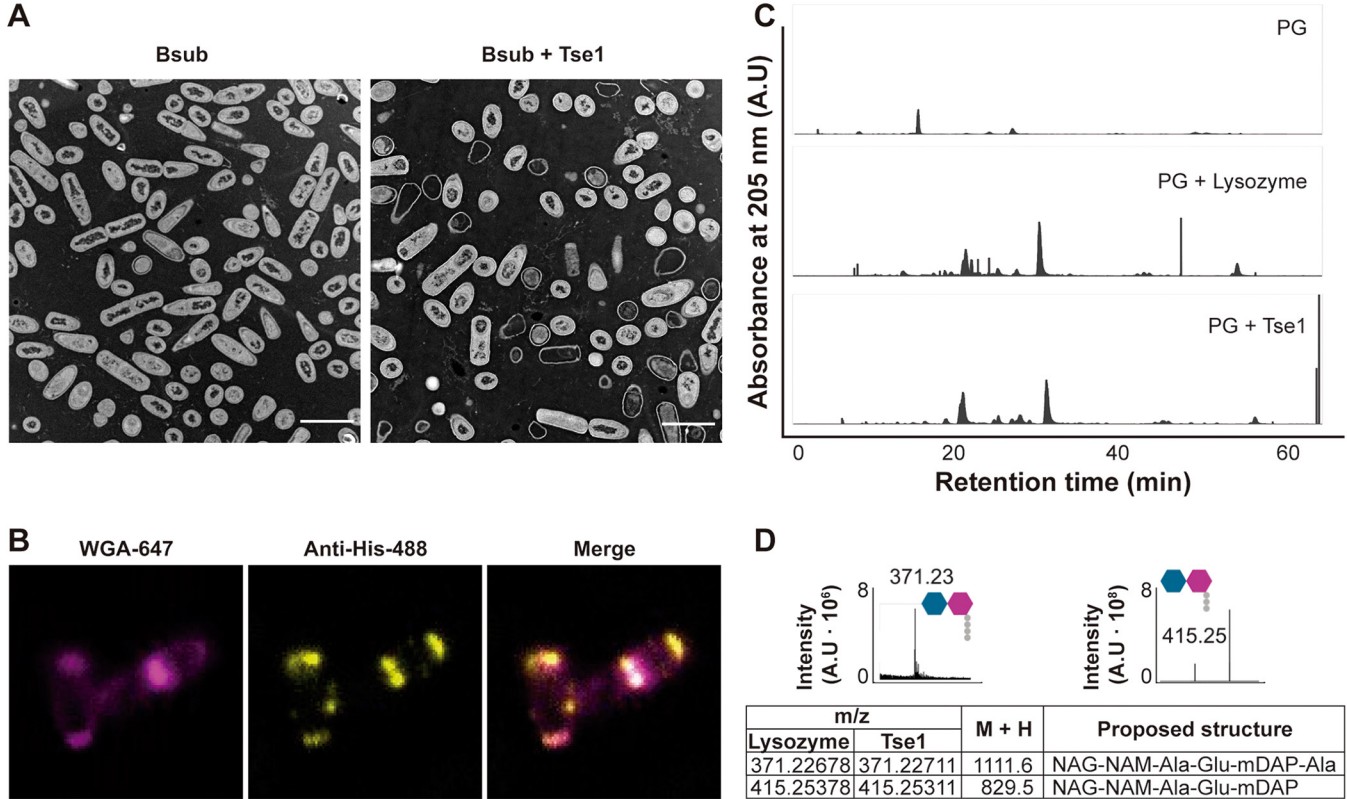

**FIG 3** Tse1 hydrolyzes *Bacillus* peptidoglycan. (A) Transmission electron micrograph of positively stained thin sections of *Bacillus* cells showing an overrepresentation of ghost cells after Tse1 treatment (Bsub + Tse1) compared with untreated cells (Bsub). After treatment with Tse1 (*n* = 85), 29.4% ghost cells were observed, compared to 0.95% ghost cells in untreated cells (*n* = 110) (*n* refers to the number of cells examined in each case). Bars, 2 μm. (B) Immunofluorescence assay of a Tse1-treated *Bacillus* bacterium immunolabeled with anti-His antibody conjugated to Alexa Fluor 488 (anti-His-488, yellow channel) and stained with WGA conjugated with Alexa Fluor 647 (WGA-647, pink channel). Protein accumulation is visible at the cell septa and poles. Images of untreated *Bacillus* cells and colocalization analysis are shown in an expanded view in Fig. 4A and B. (C) HPLC chromatograms of reduced soluble *Bacillus* peptidoglycan products obtained from digestion with lysozyme (PG + lysozyme, positive control), digestion with Tse1 (PG + Tse1), or treatment with buffer (PG [negative control]). (D) Partial HPLC chromatograms of muropeptides eluted at 55.89 min (371.23 *m/z*) and 63.65 min (415.25 *m/z*) obtained after Tse1 treatment. Peak assignments were made based on MS data and literature review. Predicted structures are shown with hexagons (NAM and NAG) and circles. The table shows the most abundant and relevant peaks found in samples treated with lysozyme or Tse1, their mass in daltons (M+H), and the proposed structures. NAG, *N*-acetylglucosamine; NAM, *N*-acetylmuramic acid; Ala, alanine; Glu, glutamic acid; mDAP, *meso*-diaminopimelic acid; A.U., arbitrary units. In all experiments, at least three biological replicates were obtained.

**$\sigma^W$ activates *Bacillus* sporulation in response to Tse1-inflicted peptidoglycan damage.** To identify the genetic pathways connecting the anatomical and functional alterations of the Bsub cell envelope with sporulation, the whole transcriptome of Bsub cells was sequenced after treatment with Tse1 for 3 h. A total of 97 genes were differentially expressed in Tse1-treated cells compared with untreated cells (Fig. 5; Data Sets S3 and S4). Consistent with the specific cytological modifications of Bsub cells (Fig. 3A), 58% of the differentially upregulated genes belonged to the $\sigma^W$ regulon, which is associated with cell wall stress, cell membrane homeostasis, and resistance to cell wall-directed antibiotics (64–67). Among the genes under positive transcriptional control by $\sigma^W$, we observed upregulation of *sasA*, which encodes the synthetase of the small alarmone guanosine tetraphosphate (ppGpp). ppGpp inhibits GMP kinase; thus, increased ppGpp leads to a decrease in the GTP pool and a corresponding increase in ATP levels, thereby enhancing transcription initiation and facilitating the autophosphorylation of KinA and KinB for sporulation phosphorelay (68–71).

To demonstrate the existence of an imbalance of the nucleotide pool after treatment with Tse1, ATP levels were quantified using a luciferase-based kit (Invitrogen). We found that ATP levels increased by a factor of three in Bsub cells treated with Tse1 compared to Bsub-untreated cells (Fig. 6A and Data Set S5). Among the five described kinases, KinA and KinB are known to sense variations in cellular levels of ATP and

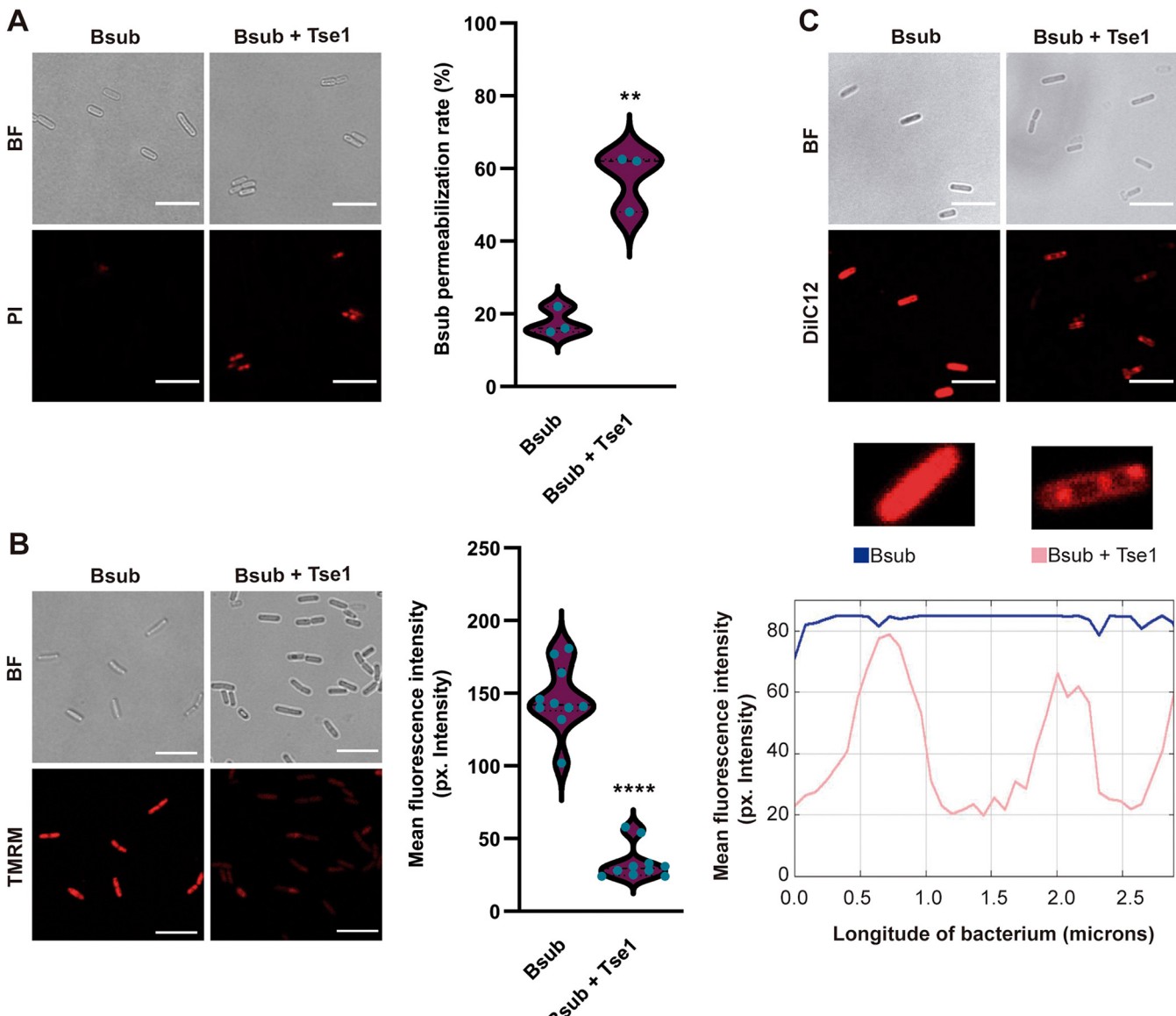

**FIG 4** Tse1 affects *Bacillus* membrane integrity. (A) CLSM of propidium iodide-stained Bsub cells revealed increased membrane permeability in Tse1-treated *Bacillus* cells (right) compared with untreated cells (left). The number of CLSM fields analyzed to calculate the number of permeabilized cells of the total of cells in the field was 3 for Bsub treated with Tse1 and for untreated Bsub. Bars, 7 $\mu$m. (B) Quantification of the fluorescence signal obtained after TMRM staining and CLSM showed a significant decrease in membrane potential in Tse1-treated Bsub cells (right) compared with that in untreated cells (left). The number of cells examined over three independent experiments to measure fluorescence intensity from TMRM staining was 10. Bars, 7 $\mu$m. (C) (Top) Staining with DilC12 and CLSM revealed higher membrane fluidity in Tse1-treated Bsub cells (right) than untreated cells (left). Bars, 7 $\mu$m. (Bottom) Representation of the plot profile of DilC12 in untreated Bsub or Tse1-treated cells showing the DilC12 distribution along a bacterium. The blue line represents the mean fluorescence intensity at each point of a single Bsub cell, and the pink line represents the mean fluorescence intensity along a single Tse1-treated Bsub cell. Experiments were biologically repeated at least three times with similar results. Statistical significance in the TMRM and PI experiments was assessed via *t* tests. **, $P < 0.01$; ****, $P < 0.001$.

activate Spo0F phosphorylation, leading to a large pool of phosphorylated Spo0A, which then ultimately controls the choice between two cell fates: sporulation and biofilm formation (Fig. 5). Strains mutant for *kinA* or *kinB*, but not for KinC and KinD kinases, were blind to the presence of WT Pchl codifying a functional T6SS or Tse1, and accordingly, their sporulation percentages were unchanged during competition with $\Delta hcp$ or $\Delta tse1$ strains (Fig. S7A).

Activation of the $\sigma^W$ regulon relies on the alleviation of the repression imposed by RsiW, the cognate anti-sigma factor of $\sigma^W$. RsiW is a transmembrane protein that under cell envelope stress is degraded by two proteases, first by PrsW and then by RasP (64, 72). Thus, we proposed that the damage inflicted on the cell wall by Tse1 might

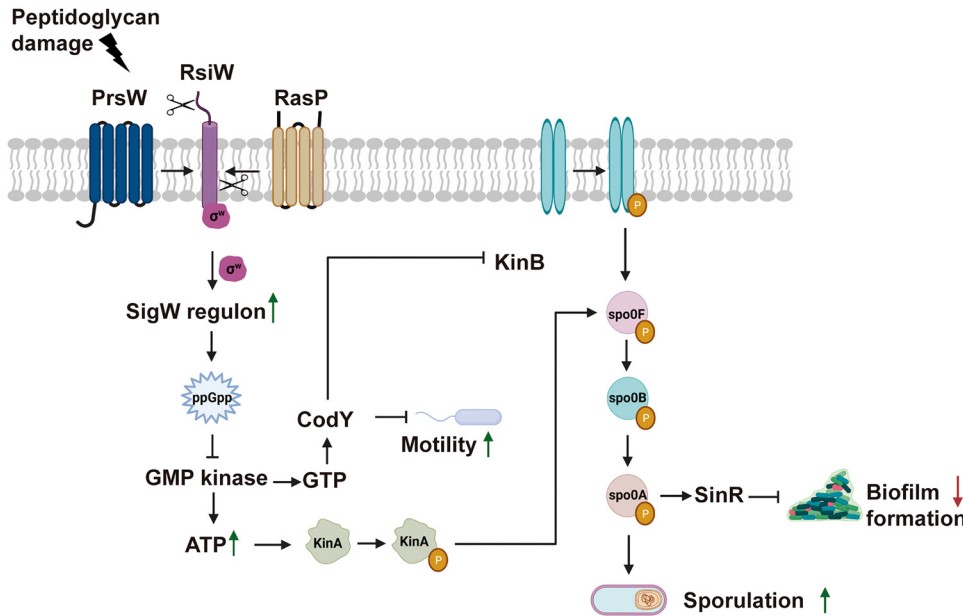

**FIG 5** Transcriptomic analysis revealed how Tse1-induced cell wall damage to Bsub activates the $\sigma^W$ regulon, which eventually triggers the KinA- and KinB-mediated phosphorelay pathway. Based on RNA-seq data, the green and red arrows show upregulated and downregulated processes, respectively. The genes that were upregulated and downregulated after treatment with Tse1 are listed in Data Sets S3 and S4, respectively.

provoke RsiW degradation, resulting in disengagement of $\sigma^W$ and activation of the genes under its positive control. The sporulation percentage of *rsiW* or *prsW* strains treated with Tse1 was not statistically different from that of untreated cells (Fig. 6B). Consistently, no increase in ATP levels was found in *rsiW* and *prsW* mutants treated with Tse1 (Fig. 6A). Confocal microscopy of Bsub cells expressing yellow fluorescent protein (YFP)-tagged RsiW (Bsub-YFP) showed signal accumulation in foci on the cell membranes of untreated cells (Fig. 6C, top, and Fig. 6D, top). Consistent with enzymatic processing of the RsiW C terminus, Tse1 treatment led to a significant decrease in fluorescent signal intensity, most likely due to the release of the C-terminally fused YFP tag (Fig. 6C, bottom, and Fig. 6D, middle). Alkaline shock (pH 10) was used to induce RsiW degradation by PrsW and RasP as positive controls (Fig. 6D, bottom) (72). Additional images of the cells shown in Fig. 6D and the quantification of the mean fluorescence intensity of analyzed cells are shown in Fig. S8A and B, respectively. In agreement with these observations, an immunoblot analysis revealed a band with an estimated molecular weight consistent with that of the RsiW-YFP translational fusion protein (Fig. 6E, blue asterisk, around 46 kDa), which was also immunoreactive to YFP antibodies in the Bsub cell fraction. Tse1 treatment eliminated this signal from the cell fraction, and a band with a molecular weight corresponding to a YFP monomer (Fig. 6E, red asterisk, around 26 kDa) and immunoreactive to YFP antibodies appeared in the cell-free supernatant. Altogether, these findings proved that the anti-sigma factor RsiW and the protease PrsW are indispensable elements of the molecular mechanism involved in sensing the peptidoglycan damage inflicted by Tse1.

## DISCUSSION

Until very recently, T6SSs were known to antagonize certain types of eukaryotic cells (e.g., mammalian and amoeboid) and Gram-negative bacteria, while they had not been linked to Gram-positive bacteria or fungal cells (7, 73–75). However, recent findings have broadened the roles of T6SSs as it is now known that the *Serratia marcescens* T6SS acts against fungal cells (22) and that the *P. chlororaphis* and *A. baumannii* T6SSs act against Gram-positive bacteria (4, 24). In this study, we identified the T6SS^PchI effector responsible for triggering Bsub sporulation and dissected the cellular and

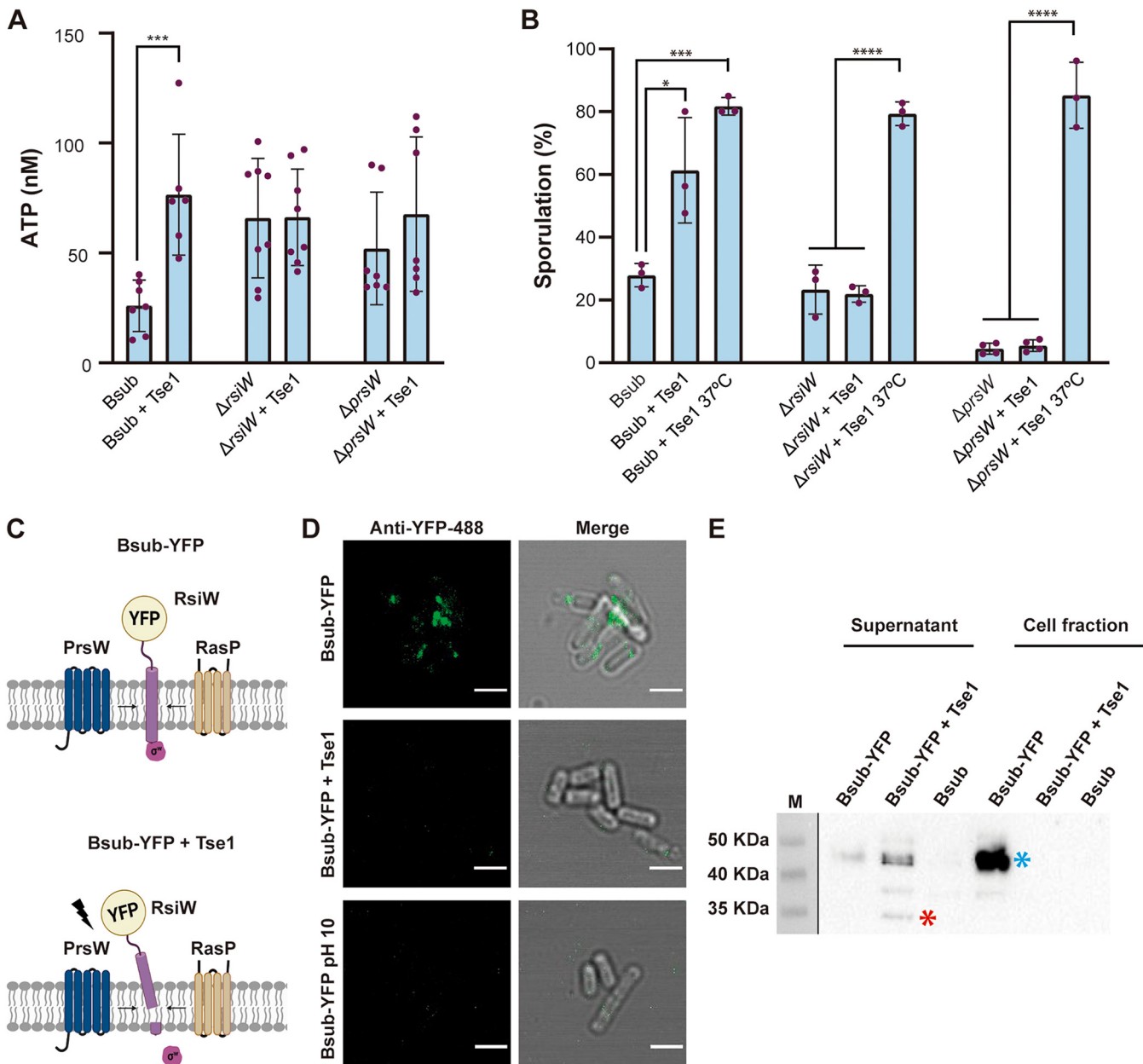

**FIG 6** *Bacillus* sporulation is mediated by $\sigma^W$ and the KinA and KinB kinases. (A) ATP levels of Bsub and *rsiW* and *prsW* strains treated with Tse1 (7 $\mu$M) and left untreated. ATP levels were calculated from luminescence signals by using a standard curve generated with standard dilutions of ATP. (B) Quantification of the percentage of sporulated Bsub, *rsiW*, and *prsW* cells after treatment with purified Tse1 showing that *rsiW* and *prsW* single mutants are blind to the presence of Tse1. Quantification of the percentage of sporulation upon treatment with Tse1 and incubation at 37°C showed that these *Bacillus* mutants (*rsiW* and *prsW*) are not intrinsically defective in sporulation (111). (C) Schematic representation of RsiW-YFP translational fusion in control Bsub cells (top) or Tse1-treated Bsub cells (bottom). (D) An immunofluorescence assay of *Bacillus* cells expressing an RsiW-YFP translational fusion using anti-YFP antibody conjugated to Alexa Fluor 488 (green channel) showed accumulation of signal in foci of untreated cells (top) and less signal intensity in Tse1-treated *Bacillus* cells (middle) and *Bacillus* cells in buffer at pH 10 (bottom). Bars, 2 $\mu$m. Additional CLSM images of immunofluorescence assay and quantification of the mean fluorescence intensity of analyzed cells for each treatment are shown in Fig. S9A and B. (E) Western blot using anti-YFP antibodies revealed an immunoreactive band of the expected size of the RsiW-YFP translational fusion in the cell fraction of untreated Bsub cells and mostly in the supernatant of Tse1-treated Bsub cells. Bsub, *Bacillus* WT; Bsub-YFP, *Bacillus* harboring the translational fusion RsiW-YFP. Gel images have been cropped and spliced for illustrative purposes. Experiments were repeated at least three times with similar results. Statistical significance was assessed via *t* tests. *, $P < 0.1$; ***, $P < 0.001$; ****, $P < 0.0001$. Uncropped images of the Western blots are shown in Data Set S9.

molecular machinery that activates sporulation in response to the damage inflicted by this effector.

The newly identified T6SS$^{PchI}$-associated effector, Tse1, partially hydrolyzes peptidoglycan, an offense that is rapidly sensed by Bsub cells, which then respond by activating sporulation. Our data indicate that Tse1 is a muramidase that hydrolyzes the peptidoglycan

backbone, similar to Tse3 from *P. aeruginosa* and Tge2 from *Pseudomonas fluorescens* (6, 52, 76), although no experimental evidence exists to support the activity of these effectors attacking Gram-positive bacteria. Given that the other reported effector which acts against Gram-positive bacteria is also a cell wall effector (24), and due to the increased difficulties of intracellularly delivering the toxins in Gram-positive bacteria, it is tempting to suggest that a high number of T6SS effectors target the cell wall. In addition to its direct activity on peptidoglycan, Tse1 also induces changes in cell membrane functionality. In particular, our *in vitro* experiments show that partial digestion on peptidoglycan caused by Tse1 compromises membrane integrity, polarization, and permeability. Such effects are not unprecedented given that three other T6SS effectors have also been found to induce similar membrane alterations: VasX, Tse4, and Ssp6 of *Vibrio cholerae*, *P. aeruginosa*, and *Serratia marcescens*, respectively (77–79).

Our findings provide new evidence supporting the versatility and variability of T6SS activity, depending especially on the strain and likely the specific competitors that cohabit in the same niche. In addition to our findings related to the effector Tse1, which appears to be predominantly present in *Pseudomonas* species (Data Set S2), we found that a strain lacking *paar* (Δ*paar*) had an active T6SS that reduced *E. coli* viability but was unable to trigger sporulation in Bsub (Fig. 2A and B). This observation is in contrast to what has been reported in *Acinetobacter baylyi* ADP1, but similar to findings from other species, including *Vibrio cholerae* and *Aeromonas dhakensis* (14, 80), where PAAR is not essential for T6SS activity. As shown in Fig. 2B, a *paar* mutant has an active T6SS able to kill *E. coli*. However, as shown in Fig. 2A, we noticed that a *paar* mutant (which encodes Tse1) is not able to trigger Bsub sporulation to a level similar to that in the WT Pchl strain. Given that *paar* deletion does not impair T6SS assembly but likely abolishes Tse1 secretion, we suggest that Tse1 is a PAAR-associated effector that requires a PAAR repeat domain protein to be targeted for secretion, thereby increasing *Bacillus* sporulation during contact with *Pseudomonas* cells (9, 81, 82).

In parallel, we propose that Tse1 and Tsi1 must form an effector-immunity pair. Tsi1 is bioinformatically predicted to be an extracellular protein (PSORTb 3.0) (83), and immunity proteins typically neutralize the toxicity of the cognate toxin by direct binding (84–86); therefore, Tsi1 must be secreted from the cell to neutralize the toxic effect of the cognate toxin Tse1 (Fig. 2D).

Finally, we have elucidated the molecular mechanism by which Tse1 triggers sporulation. Factors involved in inducing *Bacillus* sporulation during microbial interaction, such as siderophores and the compounds hadacidin and decoyinine, have been previously reported (87, 88). Based on our data, we propose a link between $\sigma^W$ activation and Bsub sporulation mediated by T6SS (Fig. 5). $\sigma^W$ belongs to the extracytoplasmic function (ECF) family of $\sigma$ factors, and it is involved in sensing cell wall and membrane stress and also in conferring resistance against antimicrobials released by competitors such as vancomycin or lantibiotics (29, 67). ECF $\sigma$ factors are two-component regulatory systems that allow bacteria to respond to extracellular conditions that disrupt cell wall homeostasis, e.g., antibiotic exposure, heat shock, or salt stress. Among the seven ECFs present in *Bacillus subtilis* ($\sigma^M$, $\sigma^V$, $\sigma^W$, $\sigma^X$, $\sigma^Y$, $\sigma^Z$, and $\sigma^{YlaC}$), $\sigma^{YlaC}$ has been directly linked to sporulation under oxidative stress (65, 89, 90). We describe how Tse1-induced peptidoglycan damage leads to $\sigma^W$ induction and activation of the sporulation cascade, thereby connecting extracellular damage sensing by RsiW with sporulation pathway activation. Based on our results, we conclude that T6SSs, and specifically Tse1, might play roles in microbial competition by modulating population dynamics.

We propose that sporulation may contribute to bacillus defense against T6SS-injected effectors. Supporting our model, *Bacillus cereus* AH187 (BcerAH) showed increased sporulation in the presence of Pchl with a functional T6SS (Fig. S9A) with no difference in cell density (Fig. S9B and C). Interestingly, a significant decrease in cell density was observed in competition experiments with the asporogenic strain *B. cereus* DSM 2302 (BcerDSM) (91), demonstrating that sporulation protects *Bacillus* against T6SS attack (Fig. S9D). According to this observation, *Bacillus* population increases in competition with the *tse1*

mutant, demonstrating that Tse1 is responsible for killing *Bacillus*. However, there is a statistical difference in the survival of *Bacillus* competing with *hcp* and *tse1* mutants. The decreased survival of *Bacillus* in the interaction with the *tse1* strain compared to *Bacillus-hcp* strain competition is suggestive of the ability of this strain to deliver additional T6SS-dependent toxins. This observation is in accordance with the data presented in Fig. 2B, which indicate that *tse1* mutant has an active T6SS able to kill *E. coli*. Thus, our data suggest that sporulation is a developmental response employed by *Bacillus* when interacting with T6SS-expressing competitors with Tse1-like effectors. This developmental process has been extensively documented due to its extreme importance in agriculture and biotechnology, and it is a striking feature of members of the genus *Bacillus* (92).

## MATERIALS AND METHODS

**Bacterial strains and culture conditions.** A complete list of the bacterial strains used in this study is shown in Data Set S6. Bacterial cultures were grown in liquid LB (lysogeny broth/Luria-Bertani broth; 1% tryptone, 0.5% yeast extract, and 0.5% NaCl) medium at 30°C (*Pseudomonas* and *Bacillus*) or 37°C (*E. coli*) with shaking on an orbital platform. The pH was adjusted to 7 prior to sterilization. When necessary, antibiotics were added to the medium at appropriate concentrations.

**Construction of *P. chlororaphis* T6SS mutants.** Chromosomal deletions were performed using the I-SceI method, based on recombination between free-ended homologous DNA sequences (93, 94). Upstream and downstream segments of homologous DNA were separately amplified and then joined to a previously digested pEMG vector using Gibson Assembly master mix (95). The oligonucleotides used are shown in Data Set S7. The resulting plasmids were then electroporated into PchI. After selection for positive clones, the pSEVA628S I-SceI expression plasmid was also electroporated, and kanamycin-sensitive clones were PCR analyzed to verify the deletions. The pSEVA628S plasmid was cured by growth without selective pressure, and its loss was confirmed by sensitivity to 60 $\mu$g/mL gentamicin and colony PCR screening.

**Construction of *B. subtilis* mutants.** All of the primers used to generate the different strains are listed Data Set S7. To build the Bsub-pDR111+tsi1 strain, *tsi* was amplified and cloned into the pDR111 overexpression vector using Gibson Assembly master mix. The resulting plasmid was transformed by natural competence into *Bsub* 168 replacing the *amyE* neutral locus. Transformants were selected via spectinomycin resistance. To build the Bsub pDR183+RsiW-YFP strain, RsiW and YFP genes were amplified by PCR and cloned into pUC19. The resulting plasmid was cloned into pDR183 by transforming *B. subtilis* 168 via its natural competence and then using positive clones as donors for transferring the constructs into Bsub via SPP1 phage transduction, replacing the *lacA* neutral locus (96).

**Bacterial competition and intoxication assays.** *Bacillus* and *Pseudomonas* strains were grown overnight in 5 mL LB before normalization to an optical density at 600 nm (OD$_{600}$) of 3.0 in 1 mL sterile distilled water. Attacker and prey strains were mixed at a 1:1 ratio, and 2.5 $\mu$L competition drops were spotted onto LB plates and incubated at 28°C for 24 h. The resulting colonies were resuspended in 1 mL of sterile distilled water and serially diluted and plated on LB. Next, *Bacillus* cells were selected by temperature at 40°C, and CFU were enumerated after 24 h. *Pseudomonas* T6SS killing assays were performed using *E. coli* DH5$\alpha$ pRL662-gfp as prey, whose plasmid confers strong constitutive GFP expression.

Overnight LB cultures of *E. coli* BL21-AI harboring pDEST17 encoding Tse1 were adjusted to an OD$_{600}$ of 0.1 and incubated at 37°C until an OD$_{600}$ of 0.6 was reached. Next, *tse1* expression was induced with 0.2% L-arabinose.

**Sporulation assays.** Spots of bacteria were resuspended in 1 mL sterile distilled water. Then, serial dilutions were made and cultured in LB solid medium for CFU counts of vegetative cells. The same serial dilutions were heated at 80°C for 10 min to kill vegetative cells and immediately cultured on LB solid medium. Plates were grown overnight at 28°C and the resulting colonies were counted to calculate the percentage of Bsub sporulation. A list of raw CFU data (total and spore population) from all figures with sporulation percentages is shown in Data Set S8.

**SDS-PAGE and immunodetection.** SDS-PAGE gels were routinely used to analyze protein samples. Precipitated proteins were resuspended in 1× Laemmli sample buffer (Bio-Rad) and heated at 100°C for 10 min. Proteins were separated via SDS-PAGE in 15% acrylamide gels and then transferred onto polyvinylidene difluoride (PVDF) membranes using the Trans-Blot Turbo transfer system (Bio-Rad) and PVDF transfer packs (Bio-Rad). For immunodetection of recombinant His-tagged Tse1, the membranes were probed with anti-His antibodies (rabbit; Sigma no. SAB1306085) used at a 1:1,000 dilution in Pierce protein-free (Tris-buffered saline [TBS]) blocking buffer. A secondary anti-rabbit IgG antibody conjugated to horseradish peroxidase (Bio-Rad no. 1706515) was used at a 1:3,000 dilution in the same buffer. The membranes were developed using the Pierce enhanced chemiluminescence (ECL) Western blotting substrate (Thermo Fisher). For immunodetection of YFP fused to RsiW, anti-YFP antibodies were used at a 1:1,000 dilution in blocking buffer, and a secondary anti-rabbit IgG antibody conjugated to horseradish peroxidase was used at a 1:3,000 dilution in the same buffer.

**Bioinformatic analysis.** *Pseudomonas* sequences were obtained from NCBI. BLASTP analyses were performed at the pseudomonas.com website. The Protein Homology Recognition Engine server (Phyre2) was used to predict structure-based homology. Protein domain predictions were performed via Pfam. The phylogenetic tree was constructed using MegaX software and the iTOL website.

All *tssB* genes from Gram-negative bacteria with characterized T6SSs were collected from the SecReT6 database. Full names and accession numbers of strains used to generate T6SS phylogenetic

tree are detailed in Data Set S1. Based on the Hcp hidden Markov model (HMM), cluster representatives were aligned using Clustal Omega (97), and a phylogenetic tree was constructed using MegaX. The phylogenetic tree was visualized using iTOL (98).

**Protein expression and purification.** To express and purify recombinant His6-tagged Tse1, *E. coli* BL21-AI was transformed with pDEST-tse1. Freshly transformed *E. coli* BL21-AI colonies harboring pDEST17-tse1 were grown in LB at 37°C overnight and reinoculated at a ratio of 1:100 in fresh LB for 2 to 3 h prior to induction of gene expression by addition of 0.2% L-arabinose and growth for 2 h at 28°C. Cells were collected via centrifugation (7,000 × *g*, 10 min, 4°C), resuspended in buffer A (50 mM Tris, 150 mM NaCl; pH 8) supplemented with 0.5 mg/mL lysozyme, 5 mM phenylmethylsulfonyl fluoride (PMSF), and 10× cell lysis reagent (Sigma), and incubated for 1 h at 37°C. Next, the cells were disrupted via sonication on ice (four times for 60 s, 80% amplitude) and passed through a 0.45-$\mu$m filter prior to protein purification via affinity chromatography using an AKTA Start fast protein liquid chromatography (FPLC) system (GE Healthcare). The lysate was loaded into a HisTrap HP 5-mL column (GE Healthcare) previously equilibrated with binding buffer (50 mM Tris, 0.5 M NaCl, 50 mM imidazole; pH 8). Protein was eluted with elution buffer (50 mM Tris, 0.5 M NaCl, 500 mM imidazole; pH 8). Next, the purified protein was loaded into a HiPrep 26/10 desalting column (GE Healthcare), and the buffer was exchanged to Tris 20 mM, NaCl 50 mM at pH 7. Finally, to obtain highly pure desalted protein via size exclusion chromatography, the protein was loaded into a HiPrep 16/60 Sephacryl S-300 HR column (GE Healthcare).

**Membrane fluidity assays.** Membrane fluidity was evaluated using the DilC12 reagent (Thermo Fisher). To evaluate regions of increased fluidity, *Bacillus* strains grown overnight on LB plates were resuspended in 1 mL of sterile distilled water followed by addition of 1 $\mu$g/mL DilC12. Benzyl alcohol (0.5%) was added to positive-control samples. Cells were incubated at 28°C for 3 h and were washed six times with sterile distilled water. Images were obtained using a Leica SP5 confocal microscope with a 63×, numerical aperture (NA) 1.3 Plan APO oil immersion objective and acquisition with excitation at 561 nm and emission detection between 576 and 640 nm. Images were processed using FIJI/ImageJ software (99). For each experiment, the laser settings, scan speed, HyD detector gain, and pinhole aperture were kept constant across all acquired images.

**Membrane potential assays.** Membrane potential was evaluated using the image-iT TMRM reagent (Invitrogen) following the manufacturer's instructions. Colonies grown at 28°C on LB plates were isolated at 24 h and resuspended in sterile distilled water. TMRM reagent was added to the bacterial suspensions at a final concentration of 100 nM, and the mixtures were incubated at 37°C for 30 min. After incubation, the cells were immediately visualized via confocal laser scanning microscopy (CLSM) with excitation at 561 and emission detection between 576 and 683 nm. The fluorescence signal from TMRM staining was quantified using CLSM, and images were processed using FIJI/ImageJ software (99).

**Membrane permeability assays.** Membrane-level cell damage was assessed via propidium iodide (PI) staining (Invitrogen). Colonies grown at 28°C on LB plates were isolated after 24 h and resuspended in sterile distilled water. Cells were collected via centrifugation and incubated with 5 $\mu$g/mL PI in the dark for 20 min. Next, fluorescence was measured with excitation at 535 nm and emission detection between 560 and 617 nm. Images were processed using FIJI/ImageJ software (99).

**Transmission electron microscopy.** Bacterial suspensions were fixed in 2% paraformaldehyde and 2.5% glutaraldehyde overnight at 4°C. After three washes in fixation mix, samples were postfixed with 1% osmium tetroxide solution for 90 min at room temperature, followed by two washes and 15 min of stepwise dehydration in an ethanol series (30%, 50%, 70%, 90%, and 100% twice). Between the 50% and 70% steps, samples were incubated en bloc in 2% uranyl acetate solution in 50% ethanol at 4°C, overnight. Following dehydration, the samples were gradually embedded in low-viscosity Spurr's resin: 1:1 resin-ethanol for 4 h, 3:1 resin-ethanol for 4 h, and pure resin overnight. The sample blocks were embedded in capsule molds containing pure resin for 72 h at 70°C. The samples were left to dry and were visualized with a FEI Talos F200X microscope.

**Peptidoglycan digestion assays.** *Bacillus* sacculus isolation was performed according to previously described methods with some modifications (100, 101). Briefly, the bacterial pellet recovered from a 50-mL exponential-phase culture was resuspended in boiling 5% SDS and incubated for 1 h. The SDS was removed via ultracentrifugation (110,000 × *g*, 10 min) and repeated water washes. Next, the *Bacillus* sacculi were sonicated (four times for 90 s, 80% amplitude) and incubated with 100 mM Tris-HCl (pH 7), 10 mg/mL RNase A, 10 mg/mL DNase, 1 M MgSO$_4$, and 100 mM CaCl$_2$ for 16 h at 37°C. To inactivate enzymes, the sacculi were treated with 5% SDS, which was then removed via ultracentrifugation (110,000 × *g*, 10 min) and repeated water washes. Next, the SDS-free pellet was resuspended in 8 M LiCl and incubated for 10 min at 37°C. The pellet was then resuspended in 100 mM EDTA and incubated for 10 min at 37°C. To remove EDTA, samples were washed with water. After ultracentrifugation, the pellet was washed with acetone. To obtain soluble muropeptides, sacculus samples were treated with lysozyme or purified Tse1 and incubated for 16 h at 37°C. A sacculus sample incubated with buffer instead of lysozyme or Tse1 served as a control. Finally, the pH of the soluble fraction was adjusted to 8.5 to 9.0 with borate buffer, and the muropeptides were reduced with freshly prepared 2 M NaBH$_4$ for 30 min. Before LC-MS analysis, the pH was adjusted to 2.0 with 25% orthophosphoric acid.

**LC-MS analysis.** Reduced muropeptides were analyzed as previously described (102), with some modifications. Briefly, sacculus samples were analyzed using an Easy nLC 1200 ultrahigh-performance liquid chromatograph (UHPLC) attached to a Q Exactive HF-X quadrupole-Orbitrap mass spectrometer (Thermo Scientific). Data were acquired using Tune 2.9 and Xcalibur 4.1.31.9. Separation was performed using an autosampler, injecting 2 $\mu$L into an analytical column (10 cm, PepMap RSLC C$_{18}$, 2 $\mu$m, 100 Å, 50 $\mu$m by 15 cm; Thermo Scientific). For chromatographic separation, solvent A (0.1% formic acid) and solvent B (acetonitrile 80% with 0.1% formic acid) were prepared. Muropeptides were separated by

using 5% B for 5 min, increasing to 20% B, further increasing to 32%, and finally increasing to 95% B over 10 min, with a constant flow rate of 0.3 mL/min. Muropeptide mass spectrometry detection was performed in positive mode with the electrospray capillary voltage at 1.5 kV at 250°C. The scan range was set to 300 to 2,000 $m/z$, and the normalized fragmentation collision energies were set at 15, 20, and 30 eV. Data were analyzed using Xcalibur Qual browser 4.2 (Thermo Fisher).

**RNA isolation and sequencing.** *Bacillus* cells were centrifuged at 4°C and then placed at −80°C for at least 30 min. For cell disruption, the cells were resuspended in RNase inhibitor buffer (20% sucrose, 10 mM Tris-HCl [pH 8], 10 mM EDTA, and 50 mM NaCl), and lysozyme (10 mg/mL) was added followed by incubation for 30 min at 37°C. After disruption, the suspensions were centrifuged, and the pellets were resuspended in TRIzol reagent (Invitrogen). Total RNA extraction was then performed as indicated by the manufacturer. DNA removal was carried out by treatment with NucleoSpin RNA Plant (Macherey–Nagel). The integrity and quality of the total RNA were assessed with an Agilent 2100 Bioanalyzer (Agilent Technologies) and via electrophoresis. rRNA was removed using the RiboZero rRNA removal (bacteria) kit from Illumina, and 100-bp single-end read libraries were prepared using a TruSeq stranded total RNA kit (Illumina). The libraries were sequenced using a NextSeq550 sequencer (Illumina). The raw reads were pre-processed with SeqTrimNext (103) using the specific NGS technology configuration parameters. This pre-processing removes low-quality, ambiguous, and low-complexity stretches, linkers, adapters, vector fragments, and contaminated sequences while keeping the longest informative parts of the reads. SeqTrimNext also discarded sequences below 25 bp. Subsequently, clean reads were aligned and annotated using the Bsub reference genome with Bowtie2 (104) in BAM files, which were then sorted and indexed using SAMtools v1.484 (105). Uniquely localized reads were used to calculate the read number value for each gene via Sam2counts (https://github.com/vsbuffalo/sam2counts). Differentially expressed genes (DEGs) were analyzed via DEgenes Hunter, which provides a combined $P$ value calculated (based on Fisher's method) using the nominal $P$ values provided by edgeR (106) and DEseq2. This combined $P$ value was adjusted using the Benjamini-Hochberg (BH) procedure (false discovery rate approach) and used to rank all the obtained DEGs. For each gene, a combination of a $P$ value of $<0.05$ and a $\log_2$ fold change of $>1$ or $<−1$ was considered the significance threshold.

**ATP measurement.** *Bacillus* cellular ATP levels were quantified using a luciferase-based kit (Invitrogen; A22066). Briefly, 200 $\mu$L of *Bacillus* cells was incubated with Tse1 (7 $\mu$M) or buffer for 3 h and then incubated with 0.5 mg/mL lysozyme for 30 min to release intracellular content. Next, cells were centrifuged (8,000 × $g$, 5 min), and the supernatants were collected to measure ATP levels. For measurement of the luminescence levels, 10 $\mu$L of each supernatant was added to 100 $\mu$L of standard solution, and light emission was quantified at 540 nm. To generate the standard curve, low-concentration ATP standard solutions were prepared and measured. Linear regression analysis was performed using GraphPad Prism version 9.

**Immunolabeling assays.** For *Bacillus* immunolabeling assays, poly-L-lysine-coated slides were used. Slides were incubated with poly-L-lysine for 1 h at room temperature and allowed to dry completely for 10 min. Cells were incubated for 24 h at 37°C with Tse1 and resuspended in PBS. After incubation for 1 h, the cells were fixed in 3% paraformaldehyde and 0.1% glutaraldehyde for 10 min. After two washes in PBS, the samples were then blocked with 3% bovine serum albumin (BSA) and permeabilized with 0.2% Triton X-100 for 1 h. The slides were then incubated with primary anti-His antibody (rabbit; 1:100 in blocking buffer) (Sigma no. SAB1306085) for 1 h. Next, the slides were washed three times with washing buffer (0.2% BSA and 0.05% Triton X-100) and incubated with YFP-conjugated secondary antibody (anti-rabbit immunoglobulin; 1:200 in blocking buffer). Finally, cells were stained prior to CLSM with Hoechst 33342 (1:1,000) and 100 $\mu$g/mL WGA-Alexa Fluor 647 conjugate (Thermo Fisher), a protein solution that binds to $N$-acetylglucosamine residues in peptidoglycan. For optimal staining, *Bacillus* cells were washed three times in 3 M KCl before incubation with WGA-Alexa Fluor 647 conjugate for 30 min. For immunolabeling assays of Bsub pDR183+RsiW-YFP, the same protocol was used but instead using anti-YFP antibody (rabbit; 1:50 in blocking buffer) (TaKaRa 632592) and YFP-488-conjugated secondary antibody (anti-rabbit immunoglobulin; 1:200 in blocking buffer). Images were processed using FIJI/ImageJ software (99).

**Statistical analysis.** Results are expressed as means and standard errors of the means (SEM). Statistical significance was assessed using analysis of variance (ANOVA) or Student's $t$ tests. All analyses were performed using GraphPad Prism version 9 or Microsoft Excel. $P$ values of $<0.05$ were considered significant.

**Data availability.** All transcriptome sequencing (RNA-seq) raw data have been submitted to Gene Expression Omnibus (GEO) and can be accessed through GEO series accession no. GSE192348.

## SUPPLEMENTAL MATERIAL

Supplemental material is available online only.
**SUPPLEMENTAL FILE 1**, XLSX file, 0.01 MB.
**SUPPLEMENTAL FILE 2**, XLSX file, 0.02 MB.
**SUPPLEMENTAL FILE 3**, XLSX file, 0.03 MB.
**SUPPLEMENTAL FILE 4**, XLSX file, 0.05 MB.
**SUPPLEMENTAL FILE 5**, XLSX file, 0.01 MB.
**SUPPLEMENTAL FILE 6**, XLSX file, 0.01 MB.
**SUPPLEMENTAL FILE 7**, XLSX file, 0.01 MB.
**SUPPLEMENTAL FILE 8**, XLSX file, 0.03 MB.
**SUPPLEMENTAL FILE 9**, XLSX file, 0.5 MB.
**SUPPLEMENTAL FILE 10**, PDF file, 0.9 MB.

## ACKNOWLEDGMENTS

We thank Saray Morales Rojas for technical support, John Pearson from the Nanoimaging Unit of Bionand for his technical support in the confocal microscopy, Josefa Gómez Maldonado from the Ultrasequencing Unit of the SCBI-UMA for RNA sequencing, and Mercedes Martín Rufián and Casimiro Cárdenas García from the Proteomic Unit of the SCAI-UMA for technical suggestions, protein sequencing, and LC-MS analysis. We are grateful to Francisco M. Cazorla (University of Málaga) for kindly providing the wild-type strain *Pseudomonas chlororaphis* PCL1606 and Patricia Bernal (University of Seville) for sharing *Pseudomonas putida* KT2440 and *P. putida* KT2440 ΔT6SS and for helpful discussion and suggestions.

This work was supported by grants from an ERC Starting Grant (BacBio 637971), Plan Nacional de I+D+i of the Ministerio de Ciencia e Innovación (PID2019-107724GB-I00), and Junta de Andalucía (P20_00479). A.I.P.L. is funded by the program FPU (FPU19/00289) and C.M.S. is funded by the program Juan de la Cierva-Incorporación (IJC2018-036923-I) and Proyectos dirigidos por jóvenes investigadores de la Universidad de Málaga (B1-2021_21).

We declare no competing interests.

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
