## [Reviewer comments · Microbiology Spectrum]

Microbiology Spectrum

Sporulation activated via σ^W protects *Bacillus* from a Tse1 peptidoglycan hydrolase T6SS effector

Alicia Pérez_Lorente, Carlos Molina-Santiago, Antonio de Vicente Moreno, and Diego Romero

Corresponding Author(s): Diego Romero, Universidad de Malaga

Review Timeline:

Submission Date:	December 7, 2022
Editorial Decision:	January 23, 2023
Revision Received:	February 8, 2023
Accepted:	February 21, 2023

Editor: Carlos Blondel

Reviewer(s): The reviewers have opted to remain anonymous.

Transaction Report:

DOI: <https://doi.org/10.1128/spectrum.05045-22>

January 23, 2023

Dr. Diego Romero
Universidad de Malaga
Microbiology
Campus de Teatinos
Málaga
Spain

Re: Spectrum05045-22 (**Sporulation activated via σ^W protects *Bacillus* from a Tse1 peptidoglycan hydrolase T6SS effector**)

Dear Dr. Diego Romero:

Link Not Available

Sincerely,

Carlos Blondel

Journals Department
Reviewer comments:

Reviewer #2 (Comments for the Author):

The research paper entitled "Sporulation activated via SigW protects *Bacillus* from a Tse1 peptidoglycan hydrolase T6SS effector" is a comprehensive work addressing the mechanisms behind the previously published observation showing that in coculture with *Pseudomonas chlororaphis* (Pchl) *Bacillus subtilis* (Bsub) induces sporulation. In this work, the authors addressed the mechanisms behind the phenomenon and provided evidence that Pchl induces Bsub sporulation by injecting T6SS toxin Tse1 into the attacked strain. This interaction with Pchl damaged the peptidoglycan and increased the membrane permeability of Bsub. Furthermore, Tse1-induced damage lead to sigW linked induction, which triggered the activation of the sporulation

cascade through RsiW dependent pathway. The authors also addressed the role of KinA and KinB and sporulation competence in this response. Overall, the work is exciting and comprehensive as it involves a variety of experimental approaches applied to shed light on the mechanism responsible for the sporulation increase in Bsub cussed by P. chlororaphis.

Although I enjoyed the work, which is very extensive and interesting, I still have some comments and questions in relation to the results/discussion sections, which I would like the authors to address. These are listed below:

1. I am wondering how the immunity protein Tsi1, which is expressed intra-cellularly from the plasmid, can protect cells from the attack by Tse1, which acts extracellularly. Do you think the immunity protein gets out? Authors should comment on this, at least in the discussion section.
2. It would be helpful if the results in Fig.S2D included a control - a strain with only a plasmid without Tse1. Also, in Fig2SB there is no indication that results are statistically different; For example, the number of spores CFUs is very similar in coculture with Pchl_r and with the mutant dtssA, however, the difference appears to be in the total number of CFUs (cell density), which is higher in coculture with the mutant than in the coculture with the WT Pchl_r. I wonder whether this difference contributes to calculations of increased sporulation frequency. From the results in Fig. S2B is difficult to conclude that there is induction of sporulation by the WT Pchl_r. Could authors comment on this in the discussion and results section? Also in lines 159-160 (results section), I miss a sentence on results presented in Fig S2B.
3. Again, in Figure S3 there is no indication of a significant difference between CFUs'? Authors should add the indications above columns and/or comment, on why there is no statistical difference in CFUs, if so.
4. Also, regarding the expression of Tsi1 immunity protein, which is expressed from the plasmid in the cell, I miss the control of the plasmid alone without the inserted immunity gene tse1. Line 198/199 - if the strain with the empty plasmid (no Tsi1) will not induce sporulation, this would be sufficient evidence.
5. Line 263/264 - The authors claim that the sporulation was not induced in kin A and kinB mutants when exposed to the Pchl_r, but both mutants have very low spo counts when in monoculture, and their sporulation was significantly induced in coculture compared to untreated controls. Why do the authors state that the two mutants in kinA or kinB were blind to induction? Am I missing something? The authors should address this point or rephrase the statements.

Comments and Suggestions for the Author:

Do NOT indicate whether the paper should be accepted or rejected.

The research paper entitled "Sporulation activated via W protects Bacillus from a Tse1 peptidoglycan hydrolase T6SS effector" is a comprehensive work addressing the mechanisms behind the previously published observation showing that in coculture with Pseudomonas chlororaphis Bacillus subtilis shows increase in sporulation. In this work the authors addressed the mechanisms of the phenomenon and provide evidence that Pchl induces Bsub sporulation by injecting toxin Tse1 through T6SS. This interaction with Pchl damages the peptidoglycan and increases membrane permeability of the attacked B. sub. Furthermore, Tse1-induced damage leads to W induction and triggers activation of the sporulation cascade though RsiW. Overall, I find the work interesting and comprehensive as it involves variety of experimental approaches applied to shed the light on the mechanism responsible for the increased sporulation of Bsub cussed by P. chlororaphis.

However, I still have some comments and questions in relation to the results/discussion sections, which I would like authors to address.

1. I am wondering how the immunity protein, which is expressed intra-cellularly can protect cells from the attack by Tse1 that acts extracellularly. Do you think the immunity protein gets out? This should be indicated in the results section and discussed.
2. The Fig.S2D is lacking a control that does not express Tse1 to prove that Tse1 is the cause pf the observed phenotype. This should be added to the figure and then figure labeled.
3. In Fig2SB there is no indication that results are statistically different, however, the number of spore's CFUs is very similar in coculture with Pchl_r and with the mutant tssA, however the difference appears to be in the total number of CFUs, which are higher in coculture with the mutant. I am wondering whether this difference contributes to calculations of increased sporulation frequency (which is the main point of the paper): From results in Fig S2B is difficult to conclude that there is induction of sporulation by the WT Pchl_r. Could authors comment on this in the discussion and results section. Lines 159-160 (results section), where I miss a sentence on results presented in Fig S2B on Pchl_r?
4. I Figure S3 there is no indication of significant difference between CFUs'? Authors should add them and add comments, why there is no statistical difference in CFUs, if so.
5. Also, regarding the expression of Tsi1 immunity protein, which is expressed from the plasmid in the cell I miss the control of the plasmid alone without inserted immunity gene tse1. Line 198/199 - a direct evidence of Tsi1 produced in B. subtilis is missing? However, if the strain with the empty plasmid (no Tsi1) will induce sporulation, this would be sufficient evidence. Again, it should be indicated if the difference in CFUs of spores with the WT Pchl_r and the mutant in coculture is significant. Authors could connect only the most relevant columns and indicate where the difference is statistically significant. Finally, I wonder how Tsi1 acts if expressed inside the cell, yet Tse1 damage is outside the cell. Authors should comment on this, at least in the discussion section.
6. Line 263/264 - The authors claim that the sporulation was not induced in kin A and kinB mutants when exposed to the Pchl_r, but both mutants have very low spo counts when in monoculture, and their sporulation was significantly induced in coculture,, compared to untreated controls. Why do authors state that they were blind to induction? Am I missing something? The authors should address this point or rephrase the statements.

Minor comments:

Line 21, the first sentence of the abstract, is not logical to me - please change it to "Within bacterial communities, community members engage in interactions employing diverse offensive and defensive tools to reach coexistence."

Line 23: change to extracellular matrix production and sporulation

Line 42: change damaging to damages

Line 94 - spo0F with a capital letter

Line 105 - the Fig. S1B indication is displaced -consider mentioning this figure in line 70, next to Fig S1A- and consider other points indicated above

Line 106 - Consider changing extracellular to extracytoplasmic

Line 116 - change KB to lowercase

Line 168 - If *P. putida* KT2440 has only two T6SS, why is the mutant KT2440 Δ T6SS referred to as a triple mutant? Is this a mistake?

The labels in Figure 2A need to be more precise in terms that one can immediately differentiate which data are associated with *P. chlororaphis* and which data with *P. putida*. Authors can add the names above.

Line 186 - Fig.S2D lacks control -consider running an extract of the strain that does not express Tse1 through the affinity column.

Line 190 - consider changing this sentence to "Tse1 injected by T6SS of the WT strain"

Line 193 - the citation Russell et al., 2011 should be changed to the number (52)

Line 262 - in Fig S8A, only data for the additional two kinases are shown but not data for KinE, although the authors write about results on three additional kinase mutants; if you have the data please add it to Fig S8A

Line 263/264 - If authors do not have the data, they should rephrase the sentence.

Line 272 - Fig. 6B The explanation is missing in the figure legend why 37 {degree sign}C was used

Lines 286 and 289 - kDa

Lines 281, 308, 309, 330 - Change references to number s

Line 361 and 362 - error: it should be written WT or decreased survival

Line 357 - change These data shown to These data show

Line 455 - change "in" to "on"

Line 495 - delete space between number and %

Line 592 - set

Line 1055 - add locus after tssB

Line 1082 - Tsi1 protein

Line 1084 - In most cases is *B. sub* - please unify this throughout the manuscript.

Fig S2A and B - add Pchl above dtssA

Fig S7 - in text Fig. 4A should be bold and not italic

Fig S8 - delete color description in brackets as it is only stated in this figure

Fig S10B - There are significant differences in sporulation % indicated in 10SA, but again no statistical difference in spore CFU counts is indicated in Fig10SB. Please explain.

Fig S9 - delete)

- In the figure legend for Figure S7 bottom panels change to right panels and the top panels to left panels

The sentence Line 323-327 is very long and hard to understand. Consider it splitting into two sentences or remove this part "as Δ hcp, Δ tssA, and Δ tse1 strains". It is repeated anyhow further down.

lines 358 -359- the sentence is unclear: " These data shown that *Bacillus* population also decreased in competition with tse1 mutant, demonstrating that Tse1 is responsible for killing *Bacillus*." If the mutant still decreased in the competition this may also suggest that there is an additional factor acting against *Bsub*. Please rephrase or explain this part in the discussion section.

Reviewer #3 (Comments for the Author):

This work identifies Tse1 as effector delivered by the T6SS system of *P. chlororaphis* playing a key role in the interaction with *B.*

subtilis and provides insights in the molecular mechanism leading to the enhanced sporulation response of the later. These findings complement previous data published by the research group and support the importance of such T6SS in interspecies interactions occurring upon contact. I really appreciate the manuscript which is well organized and with strong data in support of the findings.

I only have minor comments/concerns:

In the set-up used for co-cultivation, is there any effect on the growth of Pseudomonas, inhibition by Bacillus?

The effect of competition with Pseudomonas or mutants on sporulation of Bacillus has been determined after overnight growth (or 24h incubation) of mixed colonies on solid LB, do the authors have data supporting expression of the T6SS under these particular conditions of co-cultivation?

In the test of pure Tse1, why using specifically a 7 microM concentration? What is the dynamic of sporulation induction in these experiments, how long does it take to observe initiation of spore formation upon treatment with the pure effector?

Lysozyme-like activity of Tse1:

On what basis were identified the two peptidoglycan degradation products? Exact mass? Fragmentation in MS/MS? Both? Fig 3C is not optimal, it seems to be delays in retention times for late eluted peaks between lyso and Tse1. Where are the m/z 371 and 415 ions in these total ion chromatograms? What is the correlation between M+H and m/z ions in the table?

Fig S4: bar scale not readable as such

Fig S7: check legend for A, PI does not reflect membrane potential

Fig 6B: specify somewhere in legend that treatment at 37{degree sign}C is used as positive control and use reference to support the accuracy of the method.

Did the authors try to measure increases in the amounts of free sigW in Bacillus cells upon treatment with Tse1, if feasible?

For discussion:

Are there other exogenous compounds (than T6SS effectors) from competitors known to stimulate sporulation in B. subtilis or related species?

Info on the involvement of sigW in sensing competitors via the release of antibiotics and not enzymes such as Tse1?

Staff Comments:

Preparing Revision Guidelines

Please return the manuscript within 60 days; if you cannot complete the modification within this time period, please contact me. If you do not wish to modify the manuscript and prefer to submit it to another journal, please notify me of your decision immediately so that the manuscript may be formally withdrawn from consideration by Microbiology Spectrum.

Reviewer comments:

Reviewer #2 (Comments for the Author):

The research paper entitled "Sporulation activated via SigW protects *Bacillus* from a Tse1 peptidoglycan hydrolase T6SS effector" is a comprehensive work addressing the mechanisms behind the previously published observation showing that in coculture with *Pseudomonas chlororaphis* (Pchl) *Bacillus subtilis* (Bsub) induces sporulation. In this work, the authors addressed the mechanisms behind the phenomenon and provided evidence that Pchl induces Bsub sporulation by injecting T6SS toxin Tse1 into the attacked strain. This interaction with Pchl damaged the peptidoglycan and increased the membrane permeability of Bsub. Furthermore, Tse1-induced damage lead to sigW linked induction, which triggered the activation of the sporulation cascade through RsiW dependent pathway. The authors also addressed the role of KinA and KinB and sporulation competence in this response. Overall, the work is exciting and comprehensive as it involves a variety of experimental approaches applied to shed light on the mechanism responsible for the sporulation increase in Bsub cussed by *P. chlororaphis*.

Although I enjoyed the work, which is very extensive and interesting, I still have some comments and questions in relation to the results/discussion sections, which I would like the authors to address. These are listed below:

1. I am wondering how the immunity protein Tsi1, which is expressed intra-cellularly from the plasmid, can protect cells from the attack by Tse1, which acts extracellularly. Do you think the immunity protein gets out? Authors should comment on this, at least in the discussion section.

Thank you for the comment. Based on bioinformatic predictions (PSORTb V.3.0.3) (Yu et al., 2010), Tsi1 is secreted to the extracellular medium. Immunity proteins typically neutralise the toxicity of the cognate toxin by direct binding (Sarah Coulthurst, 2019, Srikanthasan *et al.*, 2013, Lu *et al.*, 2014), which led us to suggest that Tsi1 must be secreted in order to interact with Tse1. The specific mechanisms of inactivation are quite different along the studied toxin-antitoxin pair, therefore crystal structure studies would ideally solve the Tse1-Tsi1 interaction model.

Quote text: “In parallel, we propose that Tse1 and Tsi1 must form an effector-immunity pair. Tsi1 is bioinformatically predicted as an extracellular protein (PSORTb 3.0) (83), and immunity proteins typically neutralise the toxicity of the cognate toxin by direct binding (84-86), therefore Tsi1 must be secreted out of the cell to neutralise the toxic effect of cognate toxin Tse1 (Fig. 2D)”. See lines 339-343.

2. It would be helpful if the results in Fig.S2D included a control - a strain with only a plasmid without Tse1. Also, in Fig2SB there is no indication that results are statistically different; For example, the number of spores CFUs is very similar in coculture with Pchl_r and with the mutant dtssA, however, the difference appears to be in the total number of CFUs (cell density), which is higher in coculture with the mutant than in the coculture with the WT Pchl_r. I wonder whether this difference contributes to calculations of increased sporulation frequency. From the results in Fig. S2B is difficult to conclude that there is induction of sporulation by the WT Pchl_r. Could authors comment on this in the discussion and results section? Also in lines 159-160 (results section), I miss a sentence on results presented in Fig S2B.

We have included a Fig. S2C showing a lysate of the *E. coli* cells containing the empty plasmid, and as expected, Tse1 is not detectable in the elution fraction of the cell lysate resolved through the HisTrap HP 5mL column.

Related to FigS2B: As the reviewer correctly indicates, there are no statistical differences in terms of CFU populations. The comparison must be done using percentages of sporulation for each condition. We have included an excel (Dataset S8) with a list of raw CFUs and sporulation percentages for all the figures.

Based on your suggestions concerning these plots (FigS2B FigS3 and FigS8B), we have decided to remove them from the new version of the manuscript to avoid any confusion. We have however, maintained Dataset S8 including all the raw data.

3. Again, in Figure S3 there is no indication of a significant difference between CFUs? Authors should add the indications above columns and/or comment, on why there is no statistical difference in CFUs, if so.

Following our point stated in previous comment and based on your suggestions concerning these plots, we have decided to remove them from the new version of the manuscript to avoid confusion. We have maintained Datasets including all the raw data.

4. Also, regarding the expression of Tsi1 immunity protein, which is expressed from the plasmid in the cell, I miss the control of the plasmid alone without the inserted immunity gene *tse1*. Line 198/199 - if the strain with the empty plasmid (no Tsi1) will not induce sporulation, this would be sufficient evidence.

We have performed the competition experiments Pchl vs Bsub pDR111 empty plasmid and the results indicate no implication of the empty plasmid in the sporulation levels of Bsub, that is an increase of the sporulation percentage in the interaction with a Pchl strain expressing a functional T6SS. We have included these data in Fig2D.

5. Line 263/264 - The authors claim that the sporulation was not induced in kin A and

kinB mutants when exposed to the Pchl, but both mutants have very low spo counts when in monoculture, and their sporulation was significantly induced in coculture compared to untreated controls. Why do the authors state that the two mutants in kinA or kinB were blind to induction? Am I missing something? The authors should address this point or rephrase the statements.

The reviewer is right. We have rephrased the sentence to clarify that KinA and KinB are blind to the presence of a functional T6SS given that there is no significant difference in sporulation levels in KinA or KinB strains in competition with Pchl WT, Δhcp or $\Delta tse1$ strains.

Quote text: "Strains mutant for *kinA* or *kinB*, but not for KinC and KinD kinases, were blind to the presence of WT Pchl codifying a functional T6SS or Tse1, and accordingly, their sporulation percentages remained unchanged in the competition with Δhcp or $\Delta tse1$ strains (Fig. S7A)." See lines 262-266.

Comments and Suggestions for the Author:

Do NOT indicate whether the paper should be accepted or rejected.

The research paper entitled "Sporulation activated via σ^W protects Bacillus from a Tse1 peptidoglycan hydrolase T6SS effector" is a comprehensive work addressing the mechanisms behind the previously published observation showing that in coculture with *Pseudomonas chlororaphis* *Bacillus subtilis* shows increase in sporulation. In this work the authors addressed the mechanisms of the phenomenon and provide evidence that Pchl induces Bsub sporulation by injecting toxin Tse1 through T6SS. This interaction with Pchl damages the peptidoglycan and increases membrane permeability of the attacked B. sub. Furthermore, Tse1-induced damage leads to σ^W induction and triggers activation of the sporulation cascade through RsiW. Overall, I find the work interesting and comprehensive as it involves variety of experimental approaches applied to shed the light on the mechanism responsible for the increased sporulation of Bsub cussed by *P.chlororaphis*.

However, I still have some comments and questions in relation to the results/discussion sections, which I would like authors to address.

1. I am wondering how the immunity protein, which is expressed intra-cellularly can protect cells from the attack by Tse1 that acts extracellularly. Do you think the immunity protein gets out? This should be indicated in the results section and discussed.

Thank you for the comment. Based on bioinformatic predictions (PSORTb V.3.0.3) (Yu et al., 2010), Tsi1 is secreted to the extracellular medium. Immunity proteins typically neutralise the toxicity of the cognate toxin by direct binding (Sarah Coulthurst, 2019, Srikanthasan *et al.*, 2013, Lu *et al.*, 2014), which led us to suggest that Tsi1 must be secreted in order to interact with Tse1. The specific mechanisms of inactivation are quite different along the studied toxin-antitoxin pair, therefore crystal structure studies would ideally solve the Tse1-Tsi1 interaction model.

Quote text: "In parallel, we propose that Tse1 and Tsi1 must form an effector-immunity pair. Tsi1 is bioinformatically predicted as an extracellular protein (PSORTb 3.0) (83), and immunity proteins typically neutralise the toxicity of the cognate toxin by direct binding (84-86), therefore Tsi1 must be secreted out of the cell to neutralise the toxic effect of cognate toxin Tse1 (Fig. 2D)". See lines 339-343.

2. The Fig.S2D is lacking a control that does not express Tse1 to prove that Tse1 is the cause of the observed phenotype. This should be added to the figure and then figure labeled.

As previously mentioned, we have included a Fig. S2C showing a lysate of the *E. coli* cells containing the empty plasmid, and as expected, Tse1 is not detectable in the elution fraction of the cell lysate resolved through the HisTrap HP 5mL column.

3. In Fig2SB there is no indication that results are statistically different, however, the number of spore's CFUs is very similar in coculture with Pchl_r and with the mutant \otimes tssA, however the difference appears to be in the total number of CFUs, which are higher in coculture with the mutant. I am wondering whether this difference contributes to calculations of increased sporulation frequency (which is the main point of the paper): From results in Fig S2B is difficult to conclude that there is induction of sporulation by the WT Pchl_r. Could authors comment on this in the discussion and results section. Lines 159-160 (results section), where I miss a sentence on results presented in Fig S2B on Pchl_r?

As the reviewer correctly indicates, there are no statistical differences in terms of CFU populations. The comparison must be done using percentages of sporulation for each condition. We have included an excel (Dataset S8) with a list of raw CFUs and sporulation percentages for all the figures.

Based on your suggestions concerning these plots (FigS2B FigS3 and FigS8B), we have decided to remove them from the new version of the manuscript to avoid any confusion. We have however, maintained Dataset S8 including all the raw data.

4. I Figure S3 there is no indication of significant difference between CFUs'? Authors should add them and add comments, why there is no statistical difference in CFUs, if so.

Following our point stated in previous comment and based on your suggestions concerning these plots, we have decided to remove them from the new version of the manuscript to avoid confusion. We have maintained Datasets including all the raw data.

5. Also, regarding the expression of Tsi1 immunity protein, which is expressed from the plasmid in the cell I miss the control of the plasmid alone without inserted immunity gene tse1. Line 198/199 - a direct evidence of Tsi1 produced in *B. subtilis* is missing? However, if the strain with the empty plasmid (no Tsi1) will induce sporulation, this would be sufficient evidence. Again, it should be indicated if the difference in CFUs of spores with the WT Pchl and the mutant in coculture is significant. Authors could connect only the most relevant columns and indicate where the difference is statistically significant. Finally, I wonder how Tsi1 acts if expressed inside the cell, yet Tse1 damage is outside the cell. Authors should comment on this, at least in the discussion section.

As stated above, we have performed the competition experiments Pchl vs Bsub pDR111 empty plasmid and the results indicate no implication of the empty plasmid in the sporulation levels of Bsub, given that this strain with the empty plasmid is triggering sporulation in contact with a Pchl strain with a functional T6SS. We have included these data in Fig2D.

6. Line 263/264 - The authors claim that the sporulation was not induced in kin A and kinB mutants when exposed to the Pchl, but both mutants have very low spo counts when in monoculture, and their sporulation was significantly induced in coculture,, compared to untreated controls. Why do authors state that they were blind to induction? Am I missing something? The authors should address this point or rephrase the statements.

The reviewer is right. We have rephrased the sentence to clarify that KinA and KinB are blind to the presence of a functional T6SS given that there is no significant difference in sporulation levels in KinA and KinB when competing with Pchl WT, Δhcp or $\Delta tse1$. Quote

text: "Strains mutant for *kinA* or *kinB*, but not for KinC and KinD kinases, were blind to the presence of WT Pchl codifying a functional T6SS or Tse1, and accordingly, their sporulation percentages were unchanged during competition with Δhcp or $\Delta tse1$ strains (Fig. S7A)." See lines 262-266.

Minor comments:

Line 21, the first sentence of the abstract, is not logical to me - please change it to "Within bacterial communities, community members engage in interactions employing diverse offensive and defensive tools to reach coexistence."

Done

Line 23: change to extracellular matrix production and sporulation.

Done

Line 42: change damaging to damages

Done

Line 94 - spo0F with a capital letter.

The reviewer is right, thank you.

Line 105 - the Fig. S1B indication is displaced -consider mentioning this figure in line 70, next to Fig S1A- and consider other points indicated above

Done

Line 106 - Consider changing extracellular to extracytoplasmic

Done

Line 116 - change KB to lowercase

Done

Line 168 - If *P. putida* KT2440 has only two T6SS, why is the mutant KT2440 Δ T6SS referred to as a triple mutant? Is this a mistake?

We have rephrased the sentence to clarify that *P. putida* KT2440 possesses three T6SS, and two of them are phylogenetically close to Pchl-T6SS.

Quote text: "*P. putida* KT2440 (51), which possesses three T6SSs, two of them (K2-T6SS and K3-T6SS) phylogenetically close to Pchl, failed to induce Bsub sporulation". See line 167.

The labels in Figure 2A need to be more precise in terms that one can immediately differentiate which data are associated with *P. chlororaphis* and which data with *P. putida*. Authors can add the names above.

In Figure 2A the attacker is only *P. chlororaphis*. We believe that it would be sufficient to give the information in the figure legend instead of introducing more labels in the figure. However, we agree with the reviewer, and we have specified the *Pseudomonas* strain in each interaction within the Fig2SA where there are interactions with *P. putida* or *P. chlororaphis*.

Line 186 - Fig.S2D lacks control -consider running an extract of the strain that does not express Tse1 through the affinity column.

As mentioned above, we have included a new panel (FigS2C) with this data.

Line 190 - consider changing this sentence to "Tse1 injected by T6SS of the WT strain"

Done

Line 193 - the citation Russell et al., 2011 should be changed to the number (52)

Done

Line 262 - in Fig S8A, only data for the additional two kinases are shown but not data for KinE, although the authors write about results on three additional kinase mutants; if you have the data please add it to Fig S8A

We have rephrased the sentence.

Quote text:” Strains mutant for kinA or kinB, but not for KinC and KinD kinases, were blind to the presence of WT Pchl codifying a functional T6SS or Tse1, and accordingly, their sporulation percentages were unchanged during competition with Δhcp or $\Delta tse1$ strains (Fig. S7A)”.

Line 263/264 - If authors do not have the data, they should rephrase the sentence. Done

Line 272 - Fig. 6B The explanation is missing in the figure legend why 37 {degree sign}C was used

Done. We have added the explanation to the figure legend.

Quote text: “Quantification of the percentage of sporulation upon treatment with Tse1 and incubation at 37° show that these *Bacillus* mutants (*rsiW* and *prsW*) are not intrinsically defective in sporulation (Gauvry et al., 2021)”

Lines 286 and 289 – kDa

Done

Lines 281, 308, 309, 330 - Change references to numbers

Done, thank you.

Line 361 and 362 - error: it should be written WT or decreased survival

Done

Line 357 - change These data shown to These data show

Done

Line 455 - change "in" to "on"

Done

Line 495 - delete space between number and %

Done

Line 592 – set

Done, thank you.

Line 1055 - add locus after tssB

Done

Line 1082 - Tsi1 protein

Done

Line 1084 - In most cases is *B. sub* - please unify this throughout the manuscript.

Done

Fig S2A and B - add Pchl above dtssA

Done (see below). Pchl refers to *Pseudomonas chlororaphis*, and KT2440 to *Pseudomonas putida*.

A

Fig S7 - in text Fig. 4A should be bold and not italic

Done

Fig S8 - delete color description in brackets as it is only stated in this figure

Done, thank you.

Fig S10B - There are significant differences in sporulation % indicated in 10SA, but again no statistical difference in spore CFU counts is indicated in Fig10SB. Please explain.

Thank you. As we explained above (response to point 3) we have maintained Dataset 8. In this Figure (new Fig. S9B), total CFUs have been plotted to prove the absence of cell death due to the action of T6SS with BcerAH strain, whereas a significant decrease in cell density was observed in competition assays with BcerDSM (asporogenic strain).

B

Fig S9 – delete).

Done

- In the figure legend for Figure S7 bottom panels change to right panels and the top panels to left panels

Changed, thank you.

The sentence Line 323-327 is very long and hard to understand. Consider it splitting into

two sentences or remove this part "as Δhcp , $\Delta tssA$, and $\Delta tse1$ strains". It is repeated anyhow further down.

Done, we have removed the last part of the sentence as suggested.

lines 358 -359- the sentence is unclear: " These data shown that *Bacillus* population also decreased in competition with *tse1* mutant, demonstrating that Tse1 is responsible for killing *Bacillus*." If the mutant still decreased in the competition this may also suggest that there is an additional factor acting against *Bsub*. Please rephrase or explain this part in the discussion section.

We have rephrased the sentence for clarification.

Quote text: ". See lines 366-367.

Reviewer #3 (Comments for the Author):

This work identifies Tse1 as effector delivered by the T6SS system of *P. chlororaphis* playing a key role in the interaction with *B. subtilis* and provides insights in the molecular mechanism leading to the enhanced sporulation response of the later. These findings complement previous data published by the research group and support the importance of such T6SS in interspecies interactions occurring upon contact. I really appreciate the manuscript which is well organized and with strong data in support of the findings.

I only have minor comments/concerns:

In the set-up used for co-cultivation, is there any effect on the growth of *Pseudomonas*, inhibition by *Bacillus*?

In the previous work that gave support to this study (Molina-Santiago et al., 2019) we showed that *Pseudomonas* populations are not affected by the presence of *Bacillus* in terms of cell death. In alignment to this result, we did not observe changes in the percentage of *Pseudomonas* population in the interaction with *Bacillus* at 96 hours.

The effect of competition with *Pseudomonas* or mutants on sporulation of *Bacillus* has been determined after overnight growth (or 24h incubation) of mixed colonies on solid LB, do the authors have data supporting expression of the T6SS under these particular conditions of co-cultivation?

We have performed experiments to measure T6SS expression after 24 h of incubation and our results indicate a higher number of *Pseudomonas* cells expressing T6SS in competition with *Bacillus* than in single colony samples. See Figure below. We preferred not to include this data in the main text given that we did not pursue this line of study.

In the test of pure Tse1, why using specifically a 7 microM concentration? What is the dynamic of sporulation induction in these experiments, how long does it take to observe initiation of spore formation upon treatment with the pure effector?

We defined and tested a range of concentration around 7uM based on other manuscripts where similar T6SS-toxins with hydrolase activity have been described (Le at al., 2021 and Weber et al., 2016). We observed that 7uM was the minimal concentration of toxin with the ability to increase the percentage of sporulation.

Regarding the dynamic of sporulation induction in Bsub, we did not observe differences in the level of sporulation at 5 h. See Figure below. Changes were observed from 24 h and prevailed until the end of the experiments.

Lysozyme-like activity of Tse1: On what basis were identified the two peptidoglycan degradation products? Exact mass? Fragmentation in MS/MS? Both?

We have identified the peptidoglycan degradation products based on LC-MS and exact mass. We correlated the M+H mass with the peptidoglycan degradation products based on previous analyses of *Bacillus subtilis* 168 peptidoglycan structure and composition (Bacher et al., 2001 and Atrih et al., 1999).

Fig 3C is not optimal, it seems to be delays in retention times for late eluted peaks between lyso and Tse1. Where are the m/z 371 and 415 ions in these total ion chromatograms? What is the correlation between M+H and m/z ions in the table?

We agree with the reviewer. The delay in retention times is maintained along all the chromatograms. Samples were run at different times (see original files in Fig. S5) (we run other non-related samples in between) and, therefore, environmental conditions that might affect the LC-MS equipment changed out of our control. However, we have identified the peptidoglycan degradation products based on the exact mass and not based on retention times.

Related to the m/z 371 and 415, in FigS6 in PG+Tse1 sample, the m/z 371 is found at 55.79 min and the m/z 415 at 63.79 min. The M+H of the ion 371 is three times the m/z while the M+H of the ion 415 is twice the m/z.

Fig S4: bar scale not readable as such

We have changed the bar scale to clarify the information.

Fig S7: check legend for A, PI does not reflect membrane potential

Done

Fig 6B: specify somewhere in legend that treatment at 37°C is used as positive control and use reference to support the accuracy of the method.

Thank you. As can be seen in our response to reviewer 1, we have added the next information to the legend. Quote text: “Quantification of the percentage of sporulation upon treatment with Tse1 and incubation at 37° show that these *Bacillus* mutants (*rsiW* and *prsW*) are not intrinsically defective in sporulation (108) (Gauvry et al., 2021)”.

Did the authors try to measure increases in the amounts of free sigW in *Bacillus* cells upon treatment with Tse1, if feasible?

As we stated in this work, Tse1 is sensed by RsiW-SigW, a two-component system that when cells experience an appropriate envelope stress the anti- σ factor is degraded, which leads to release of the σ factor and activation of the regulon (Helmann et al., 2016). Based on our RNAseq analysis, we have not observed an increase in the expression level of SigW, supporting the release of SigW from RsiW as the most reasonable explanation. In accordance with this suggestion, immunocytochemistry assays showed less fluorescent signal related to RsiW upon addition of the toxin Tse1 (Figs. 6C and 6D).

For discussion:

Are there other exogenous compounds (than T6SS effectors) from competitors known to stimulate sporulation in *B. subtilis* or related species?

Thank you for your suggestion, we have included this information in the discussion section.

Quote text: “Factors involved in induction of sporulation of *Bacillus* during microbial interaction, such as siderophores and the compounds hadaicin or decoynine have been previously reported (84,85)”. See lines 344-346.

Info on the involvement of sigW in sensing competitors via the release of antibiotics and not enzymes such as Tse1?

We have included information and references related to sigW sensing antimicrobial compounds and lantibiotics in discussion.

Quote text: “ σ^W belongs to the extracytoplasmic function (ECF) family of σ factors and it is involved in sensing cell wall and membrane stress, and also in conferring resistance against antimicrobials released by competitors such as vancomycin or lantibiotics (29,67)”. See lines 347-351

References

PSORTb v3.0: N.Y. Yu, J.R. Wagner, M.R. Laird, G. Melli, S. Rey, R. Lo, P. Dao, S.C. Sahinalp, M. Ester, L.J. Foster, F.S.L. Brinkman (2010) PSORTb 3.0: Improved protein subcellular localization prediction with refined localization subcategories and predictive capabilities for all prokaryotes, *Bioinformatics* 26(13):1608-1615

Coulthurst S. The Type VI secretion system: A versatile bacterial weapon. *Microbiol (United Kingdom)*. 2019;165(5):503–15.

Srikannathasan V, English G, Bui NK, Trunk K, O'Rourke PEF, Rao VA, et al. Structural basis for type VI secreted peptidoglycan dl-endopeptidase function, specificity and neutralization in *Serratia marcescens*. *Acta Crystallogr Sect D Biol Crystallogr*. 2013;69(12):2468–82.

Lu D, Shang G, Zhang H, Yu Q, Cong X, Yuan J, et al. Structural insights into the T6SS effector protein Tse3 and the Tse3-Tsi3 complex from *Pseudomonas aeruginosa* reveal a calcium-dependent membrane-binding mechanism. *Mol Microbiol*. 2014;92(5):1092–112.

Gauvry E, Mathot AG, Couvert O, Leguérinel I, Coroller L. Effects of temperature, pH and water activity on the growth and the sporulation abilities of *Bacillus subtilis* BSB1. *Int J Food Microbiol*. 2021;337(October 2019).

Molina-Santiago, Pearson JR, Navarro Y, Berlanga-Clavero MV, Caraballo-Rodriguez AM, Petras D, et al. The extracellular matrix protects *Bacillus subtilis* colonies from *Pseudomonas* invasion and modulates plant co-colonization. *Nat Commun*. 2019;10(1).

Le NH, Pinedo V, Lopez J, Cava F, Feldman MF. Killing of Gram-negative and Gram-positive bacteria by a bifunctional cell wall-targeting T6SS effector. *Proc Natl Acad Sci U S A*. 2021;118(40):6–11.

Weber BS, Hennon SW, Wright MS, Scott NE, de Berardinis V, Foster LJ, et al. Genetic dissection of the type VI secretion system in *Acinetobacter* and identification of a novel peptidoglycan hydrolase, TagX, required for its biogenesis. *MBio*. 2016;7(5):1–17.

Atrih A, Bacher G, Allmaier G, Williamson MP, Foster SJ. Analysis of peptidoglycan structure from vegetative cells of *Bacillus subtilis* 168 and role of PBP 5 in peptidoglycan maturation. *J Bacteriol*. 1999;181(13):3956–66.

Bacher G, Körner R, Atrih A, Foster SJ, Roepstorff P, Allmaier G. Negative and positive ion matrix-assisted laser desorption/ionization time-of-flight mass spectrometry and positive ion nano-electrospray ionization quadrupole ion trap mass spectrometry of peptidoglycan fragments isolated from various *Bacillus* species. *J Mass Spectrom*. 2001;36(2):124–39.

Helmann JD. *Bacillus subtilis* extracytoplasmic function (ECF) sigma factors and defense of the cell envelope. *Curr Opin Microbiol* [Internet]. 2016;30:122–32. Available from: <http://dx.doi.org/10.1016/j.mib.2016.02.002>

Mitani, T., Heinze, J. E. & Freese E. Induction of sporulation in *Bacillus subtilis* by decoynine or hadacidin. *Biochem Biophys Res Commun*. 1977;77(1):1.

Grandchamp GM, Caro L, Shank EA. Pirated siderophores promote sporulation in *Bacillus subtilis*. *Appl Environ Microbiol*. 2017;83(10).

Helmann JD. Deciphering a complex genetic regulatory network: The *Bacillus subtilis* σ^W protein and intrinsic resistance to antimicrobial compounds. *Sci Prog*. 2007;89 PART 3(4/-):243–66.

Cao M, Wang T, Ye R, Helmann JD. Antibiotics that inhibit cell wall biosynthesis induce expression of the *Bacillus subtilis* σ^W and σ^M regulons. *Mol Microbiol*. 2002;45(5):1267–76.

February 21, 2023

Dr. Diego Romero
Universidad de Malaga
Microbiology
Campus de Teatinos
Málaga
Spain

Re: Spectrum05045-22R1 (**Sporulation activated via σ^W protects *Bacillus* from a Tse1 peptidoglycan hydrolase T6SS effector**)

Dear Dr. Diego Romero:

Your manuscript has been accepted, and I am forwarding it to the ASM Journals Department for publication. You will be notified when your proofs are ready to be viewed.

Sincerely,

Carlos Blondel
Editor, Microbiology Spectrum
